# Time-resolved oxidative signal convergence across the algae–embryophyte divide

Tim P. Rieseberg [1,13] ✉, Armin Dadras [1,13], Tatyana Darienko [1,2], Sina Post[3], Cornelia Herrfurth [3,4], Janine M. R. Fürst-Jansen [1], Nils Hohnhorst[1], Romy Petroll [5], Stefan A. Rensing [6], Thomas Pröschold [1,7], Sophie de Vries [1,12], Iker Irisarri [1,8,9,10], Ivo Feussner [3,4,11] & Jan de Vries [1,8,12] ✉

The earliest land plants faced a significant challenge in adapting to environmental stressors. Stress on land is unique in its dynamics, entailing swift and drastic changes in light and temperature. While we know that land plants share with their closest streptophyte algal relatives key components of the genetic makeup for dynamic stress responses, their concerted action is little understood. Here, we combine time-course stress profiling using photophysiology, transcriptomics on 2.7 Tbp of data, and metabolite profiling analyses on 270 distinct samples, to study stress kinetics across three 600-million-year-divergent streptophytes. Through co-expression analysis and Granger causal inference we predict a gene regulatory network that retraces a web of ancient signal convergences at ethylene signaling components, osmosensors, and chains of major kinases. These kinase hubs already integrated diverse environmental inputs since before the dawn of plants on land.

Earth's surface teems with photosynthesizing life. Biodiverse cyanobacteria and algae form green biofilms on rocks and tree barks, and lichens thrive on the bleakest mountaintops. All of this is however dwarfed by the lineage that conquered land globally: the land plants (embryophytes)[1]. Together with streptophyte algae, land plants belong to the streptophytes[2]. Phylogenomic analyses established that the Zygnematophyceae are the closest streptophyte algal relatives of land plants[2–4] and comparative genomics have ushered in major progress in establishing a shared catalog of genes for key traits between streptophyte algae and land plants[5–10]. Yet, we are only beginning to understand how these genes might have been used in a functional advantage at the time of the conquest of land[11]. Several synergistic properties that hinge on a complex genetic chassis have shaped the plants that conquered land[12], including multicellular development[13,14],

[1]University of Göttingen, Institute of Microbiology and Genetics, Department of Applied Bioinformatics, Goldschmidtstr. 1, 37077 Göttingen, Germany. [2]University of Göttingen, Albrecht Haller Institute of Plant Science, Experimental Phycology and Culture Collection of Algae at Göttingen University (EPSAG), Nikolausberger Weg 18, 37073 Göttingen, Germany. [3]University of Göttingen, Albrecht Haller Institute of Plant Science, Department of Plant Biochemistry, Justus-von-Liebig-Weg, 37077 Göttingen, Germany. [4]University of Göttingen, Goettingen Center for Molecular Biosciences (GZMB), Service Unit for Goettingen Metabolomics and Lipidomics, Justus-von-Liebig Weg 11, 37077 Göttingen, Germany. [5]Department of Algal Development and Evolution, Max Planck Institute for Biology Tübingen, Tübingen, Germany. [6]University of Freiburg, Centre for Biological Signalling Studies (BIOSS), Freiburg, Germany. [7]University of Innsbruck, Research Department for Limnology, 5310 Mondsee, Austria. [8]University of Göttingen, Campus Institute Data Science (CIDAS), Goldschmidtstr. 1, 37077 Göttingen, Germany. [9]Section Phylogenomics, Centre for Molecular Biodiversity Research, Leibniz Institute for the Analysis of Biodiversity Change (LIB), Museum of Nature, Hamburg, Martin-Luther-King Platz 3, 20146 Hamburg, Germany. [10]Museo Nacional de Ciencias Naturales (MNCN-CSIC), Department of Biodiversity and Evolutionary Biology, José Gutiérrez Abascal 2, 28006 Madrid, Spain. [11]University of Göttingen, Göttingen Center for Molecular Biosciences (GZMB), Department of Plant Biochemistry, Justus- von-Liebig Weg 11, 37077 Göttingen, Germany. [12]University of Göttingen, Göttingen Center for Molecular Biosciences (GZMB), Department of Applied Bioinformatics, Goldschmidtstr. 1, 37077 Göttingen, Germany. [13]These authors contributed equally: Tim P. Rieseberg, Armin Dadras. ✉e-mail: timphilipp.rieseberg@uni-goettingen.de; devries.jan@uni-goettingen.de

propagation[15], symbiosis[16,17], and stress response[18]. In case of the latter, the earliest land plants had to overcome a diverse range of stressors to which modern land plants dynamically respond by adjusting their growth and physiology[19]. One of the hallmarks of abiotic stress on land in contrast to water is its more dynamic nature: life on land involves rapid and drastic shifts in temperature, light or water availability[18]. We focus on two terrestrial stressors: strongly fluctuating temperatures (cold and heat stress) and light conditions (high light stress and recovery).

Terrestrial stressors impact plant and algal physiology especially through the generation of reactive oxygen species (ROS) in the plastid; the plastid acts as a signaling hub upon environmental challenge[20–22]. Carotenoids are integral in the oxidative stress mitigation networks of Chloroplastida and found in nearly any photosynthetic organism[23,24]. By quenching oxidative stress of different nature, oxidative breakdown products of the polyene backbone called apocarotenoids are a consequence[25,26]. Apocarotenoids act as signals in attuning plant and plastid physiology to stress[27–31]. The diversity of apocarotenoids is vast, including land plant hormones like abscisic acid (ABA)[32,33] and strigolactones[34] but also small volatiles like β-ionone (β-IO) and β-cyclocitral (β-CC) with a growing number of recognized functions[27–31,35]. The involvement of β-CC in high light stress response was confirmed by several studies in *Arabidopsis thaliana*[27,29] and some data also suggests a role in retrograde signaling[36,37]. A physicochemical consequence of the elevation of atmospheric oxygen levels due to plant terrestrialization and radiation[38] might have been higher rates of apocarotenogenesis—even independent of the evolution of carotenoid-cleaving enzymes. The utilization of signals derived from carotenoids in the first land plants is hence plausible and adaptive.

Here, we studied the integration of apocarotenoid signals and oxidative stress networks in three genome-sequenced non-vascular streptophytes: the algae *Zygnema circumcarinatum*[10] and *Mesotaenium endlicherianum*[7,11], and the land plant *Physcomitrium patens*[39]. Of data on pigment profiles and photophysiology correlations were calculated with time-resolved global gene expression profiles from 270 biological samples in total. Using gene co-expression and gene regulatory network inference, we retrace a web of ancient kinase hubs where environmental and apocarotenoid signals converged already in the last common ancestor of embryophytes and algae.

## Results and discussion

### A comparative framework for stress dynamics across 600 million years of streptophyte evolution

We worked with three genome-sequenced streptophytes that are 600-million-years-divergent (Fig. 1a) under comparable growth conditions (see Methods; briefly, 20/25 °C, 80–100 μmol photons m$^{-2}$ s$^{-1}$): two representatives of the diverse and species-rich[40] closest algal relatives of land plants, the filamentous alga *Zygnema circumcarinatum*[10] SAG 698-1b (henceforth *Zc*) and the unicellular alga *Mesotaenium endlicherianum*[7,11] SAG 12.97 (*Me*) and the bryophyte model *Physcomitrium patens*[39] strain Gransden 2004 (*Pp*). After growth to apt density, we challenged the organisms by shifting them to (i) high light stress (HL) at 10x growth light intensity (i.e., 800–1050 μmol photons m$^{-2}$ s$^{-1}$) for 6 h followed by (ii) 4 h of recovery at standard growth conditions, and 24 h of (iii) low temperature at 12 °C colder and (iv) high temperature at 12 °C warmer than standard growth conditions; all experimentation was repeated in three successive biological replicates (Fig. 1b). To track the physiological response of the three streptophytes, we measured their relative photosynthetic yield (Fq′/Fm′) using pulse-amplitude-modulation (PAM) fluorometry (Fig. 1c–e) at 13 time points (Fig. 1b).

On balance, responses upon stress were well conserved in the three studied organisms but acted in slightly differing kinetics (Fig. 1c). HL brought the swiftest changes in Fq′/Fm′ (Fig. 1e), with an average drop by 0.49 ± 0.08 at 30 min (Fig. 1c). While *Me* and *Pp* recovered

progressively after HL, *Zc* gusted to the initial values but recovered later (Fig. 1c). In all three studied species, HL caused a stronger and faster response in Fq′/Fm′ than temperature stress (Fig. 1c). In *Zc* and *Me* the stressors additionally induced morphological alterations (Fig. 1f)—in both causing significant changes in chloroplast shape upon HL and heat as well as LD accumulation upon cold (Fig. 1g); in *Zc*, heat and HL further induced pigment alterations and plasmolysis (Fig. 1g).

Overall, our setup captures quantifiable changes in physiology that follow a stress kinetic. We next turned to the molecular consequences.

### Global transcriptomics bear out divergent time-course dynamics

To understand how the temporal and amplitudinal differences in physiology relate to genetic changes, we used global gene expression analysis. We generated RNA-seq data on samples from 29 different conditions in biological triplicates (sextuplicates for t$_0$) across all three species, yielding a total of 270 biological samples. All sequencing was performed on the NovaSeq6000 platform (Illumina) using paired-end 150 bp reads, yielding 2.8 Tbp (2,806,144,787,400 bp), of which 2.7 Tbp were retained after quality filtering (2,710,780,974,000 bp). We mapped the data onto the respective genomes based on current gene models[10,11,39] with an average alignment rate of 83.39%, 67.39%, and 92.58 % for *Me*, *Zc*, and *Pp*, respectively. A principal component analysis (PCA) of gene expression data recovered a clear separation between stress and control conditions (Fig. 2a). Along PC1 (describing between 26.9% to 44% of the variance; Fig. 2a) heat showed the swiftest separation upon temporal progression in all three streptophytes followed by HL for the two algae (Fig. 2b); recovery caused time-successive rapprochement to control conditions (Fig. 2a). Treatment with cold generally induced less differences from control conditions and resembled characteristics of a resting state despite slight fluctuations likely due to circadian rhythms.

To pinpoint the functions that underpin the temporal dynamics of stress response, we performed differential gene expression analysis of each treatment to its respective control. We carried out 22 comparisons for each species (Supplementary Fig. 1a), yielding up to more than 3000 significantly differentially expressed genes (Supplementary Fig. 1b), and performed GO term enrichment across time and treatments (Fig. 2c); irrespective of the treatment, more than 50% of the differentially expressed genes fall into the hierarchical orthogroups (HOG) shared by all species (Supplementary Fig. 2a). HL induces rapid responses whereas responses to cold were more delayed except in *Me*. In general, *Me* showed the highest responsiveness towards the studied environmental cues (Supplementary Fig. 1b). Further, on balance, *Zc* showed the most delayed response (Fig. 2c)—even in the recovery (Supplementary Fig. 2b); here, *Pp* appeared to recover the fastest (Supplementary Fig. 2), consistent with its physiological profile (Fig. 1c). This subdued response of *Zc* (Fig. 2b, c; and in the following) is likely due to its natural ecophysiology and growth, which allows the formation of highly resilient algal mats that even withstand the harsh environments of the High Arctic[41].

Recurrent themes corroborated previous findings[11,42]: among the responses, light quality signaling, response to ROS, and photosynthesis-associated processes stuck out (Supplementary Fig. 3; Supplementary Data 1). In *Me* calcium signaling and kinases featured prominently in the temperature stresses (Fig. 2c). Overall, the gene expression data align with the physiological data. We next turned to a physiological stress mitigation mechanism that generates signals: carotenoid metabolism and apocarotenoid signals.

### Xanthophyll cycle and apocarotenogenesis were the most dynamic in *Mesotaenium*

Under stress, carotenoid levels adjust and yield apocarotenoid signals through oxidative cleavage[23] (Fig. 3a). To assess their profiles, we

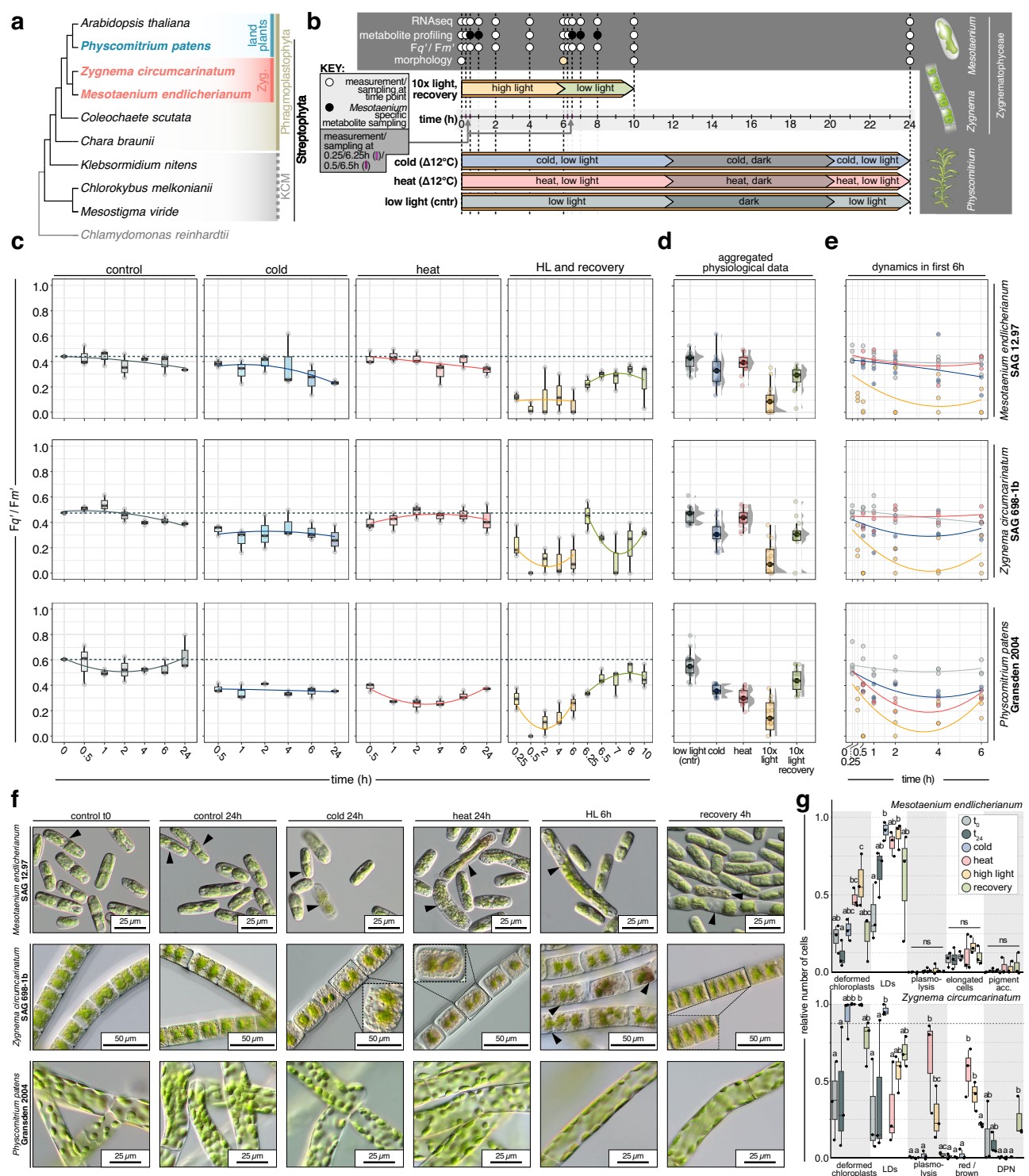

combined RP-C$_{30}$-HPLC-UV-Vis (carotenoids and chlorophylls) with HS-SPME-GC-MS (volatile apocarotenoids; Fig. 3b, c, e) and investigated transcript levels of relevant genes (Fig. 3d). In general, pigment fluctuations due to stress exposure seem more similar in the two studied multicellular species *Zc* and *Pp* than in the unicellular *Me* (Fig. 3b, c, e); for example, the de-epoxidation state changed after 15 min of HL by 1.98-fold in *Me*, 1.09-fold in *Zc*, and 1.35-fold in *Pp*. Changes in the relative xanthophyll pool size were ±20% upon 2 h of treatment by any stressor relative to t$_0$ (Fig. 3c), suggesting that the overall response is mainly determined by the altered physicochemical stress conditions. Transcript levels of the enzyme-coding genes (e.g., *PDS*, *CRTISO*,

*CYP97A*) likely underpinning (apo)carotenogenesis were down-regulated, suggesting a feedback loop: for example, after 6 h, *PDS* was downregulated—relative to control—in *Me* 44-fold (heat) and 13.5-fold (HL) (BH-corrected $P = 1.28 \times 10^{-17}$ and $1.52 \times 10^{-11}$), and in *Zc* 2.4-fold (heat) and 1.6-fold (HL) (BH-corrected $P = 7.64 \times 10^{-6}$ and 0.016); in *Pp* PDS expression did not change or was upregulated by 3.2-fold under cold (BH-corrected $P = 8.68 \times 10^{-12}$; Fig. 3d). The pigment pools of *Me* and *Pp* are of similar sizes (compared to *Pp*, t$_0$ 95% and 100%) while *Zc* has a much smaller pool (13%; Fig. 3c)—but it could be that the weight of mucilage skews the values for *Zc*. The pigment pool of *Me* was the most dynamic, especially with regard to the ratios of the xanthophylls

**Fig. 1 | A time-series setup for stress kinetics in three 600-million-year-divergent streptophytes. a** Cladogram of Streptophyta; Phragmoplastophyta studied herein are highlighted in bold. **b** Summary of the stress experiment time grids and the respective investigations including RNAseq, metabolite profiling, photophysiology, and morphology. *Me*-specific metabolite profiling is indicated by black dots. For high light stress experiments, morphology was investigated at maximum stress (yellow dot) and maximum recovery. **c** Boxplots of relative quantum yield of PSII (Fq′/Fm′) during stress exposure and control, categorized by time scale from three independent biological replicates. **d** Boxplots of aggregated relative quantum yield of PSII (Fq′/Fm′) due to stress exposure and in control; data are aggregated based on the three biological replicates shown in **c. e** Scatterplots with Loess fit of relative quantum yield of PSII (Fq′/Fm′) during stress exposure and control in the first six hours of the experiment, absolute time scale. **f** Morphological observations before and after stress exposure with time points corresponding to **b**; arrowheads and zoom-ins mark notable observations that also went into the quantification. **g** Quantification of observed morphological effects; statistics was performed using a Kruskal–Wallis followed by a two-sided Dunn's test for multiple comparisons. LDs, accumulated lipid droplets/small droplets; red/brown, overall color change of cell to a more red-brown shade; DPN, droplets accumulated around the nucleus. In all boxplots, horizontal lines within the boxes represent the median ($Q_2$); the box depicts the interquartile range (IQR) from the 25th ($Q_1$) to the 75th percentile ($Q_3$) while the whiskers extend to minimum or maximum values in the data determined by $Q_1 - 1.5 \times IQR$ to $Q_3 + 1.5 \times IQR$ (data points outside are defined as outliers). In **d**, a gray half-eye density plot next to the boxes visualizes the data distribution. Quantifications of morphological changes were done on three independent biological replicates per species and treatment; for each replicate, six to ten images where captured, wherein four to eight cells were evaluated. The *P* values of significant changes were: *Me* deformed chloroplasts, 0.00578 and 0.00279 in heat and high light versus 24 h; *Me* LDs, 0.00735 and 0.01280 in cold and high light versus t0; *Zc* deformed chloroplasts, 0.01466 for heat and HL versus 24 h; *Zc* LDs, 0.02386 for cold versus 24 h; *Zc* plasmolysis, 0.00422 and 0.02209 for heat and HL ersus 24 h; *Zc* red/brown, 0.00354 and 0.01166 for heat and HL versus 24 h; *Zc* DPN, 0.00884 for all stressors versus revovery. All *P* values can be found in the source data.

antheraxanthin (A) + zeaxanthin (Z) to violaxanthin (V) + A + Z (AZ/VAZ) (Fig. 3c, e). Further, the ratio of β-carotene to the apocarotenoids β-CC, β-IO, and dihydroactinidiolide (DHA) had changed after 2 h of treatment most pronouncedly upon HL, down to 0.36, 0.59, and 0.31 for *Me, Zc*, and *Pp* compared to 1.05, 0.59, and 0.41 in cold and 0.35, 0.9, and 0.65 upon heat stress (Fig. 3b, e). As expected, there was no direct correlation between changes in apocarotenoid levels and transcript levels, because apocarotenoids are also formed by non-enzymatic cleavage. The non-enzymatic cleavage reaction is favored under elevated temperatures as well as HL, and we observe decreasing transcript levels: in *Me* and *Pp*, CCD1 showed the most dynamic response in the first 2 h of treatment, decreasing 3.3- and 5.5-fold in *Me* upon heat and HL stress (BH-corrected $P = 7.63 \times 10^{-7}$ and $8.05 \times 10^{-13}$) and 3.9-fold in *Pp* upon heat stress (BH-corrected $P = 1.56 \times 10^{-15}$). Me*CCD1* constantly rose during the first 6 h cold stress up to 6.4-fold (BH-corrected $P = 1.12 \times 10^{-11}$), while β-CC, β-IO levels reached their peak at 4 h cold. β-IO appears to be swiftly converted to DHA in a non-enzymatic reaction[26,27]. DHA has also shown a stronger effect on oxidative stress-responsive genes in earlier studies[27,43]. Overall, we recovered the presence of well-known apocarotenoid signals with the most pronounced stress kinetics in *Mesotaenium*.

## Biological programs correlate with the environmental triggers and pigment profiles

Each organism reacted to the environmental cues by the altered expression of up to thousands of genes. To understand their cooperative action, we clustered all 13,125, 9445, and 15,778 genes that passed the expression cutoff in *Me, Zc*, and *Pp* into 27, 34, and 29 clusters respectively, to which we refer by color (Fig. 4a). We then asked the questions of how expression patterns of the genes assigned to these clusters (i) correlate with the environmental cues and apocarotenoid levels (Fig. 4b) and (ii) are similar across species (Fig. 4c, d). For the latter, we worked with Orthofinder's hierarchical orthogroups (HOGs) and calculated Jaccard distances (Fig. 4c, d).

The gene clusters consist of meaningful biological cohorts, such as *Me*darkred and *Pp*darkgrey that are Jaccard-similar (Fig. 4d) and enriched in ribosomal GO-terms, and ribosome component-coding genes that are hubs in these networks (Supplementary Figs. 4 and 5). Physiologically coherent behaviors were recovered. For example, cluster *Me*yellow shows negative correlation with temperature ($r = -0.73$, $P = 10^{-13}$) and the ratios of AZ/VAZ ($r = -0.52$, $P = 2 \times 10^{-6}$), with the Jaccard similar cluster *Pp*yellow showing a negative correlation with temperature ($r = -0.67$, $P = 10^{-10}$) and similar terms and hub genes associated with chloroplast physiology (Fig. 4c; Supplementary Fig. 5) which was as in previous work[11] a recurrent theme (Fig. 4k); in contrast, for example, *Me*darkorange has weak correlation with temperature ($r = 0.35$, $P = 0.003$) and light intensity ($r = 0.33$, $P = 0.006$;

Supplementary Fig. 5). Clusters of *Zc* seldom show correlations with the tested environmental factors but this is likely due to *Zc*'s subdued responses. Overall, the clearest correlations are with temperature, where highly Jaccard-similar clusters (Fig. 4c, d) such as *Me*brown (Fig. 4e), *Pp*brown (Fig. 4f), *Zc*brown (Supplementary Fig. 5), and *Me*Turquoise (Fig. 4g) show strong positive correlations ($r = 0.85$, $P = 5 \times 10^{-23}$; $r = 0.64$, $P = 10^{-9}$; $r = 0.74$, $P = 2 \times 10^{-13}$; and $r = 0.39$, $P = 7 \times 10^{-4}$; Fig. 4b) and similar terms across species associated with protein homeostasis.

Several clusters reflect signaling processes, including genes for kinases and HISTIDINE KINASE (AHK) hubs (Fig. 4f, h, i; Supplementary Fig. 5), calcium and Ca²⁺-DEPENDENT KINASE (CDPK) hubs (Fig. 4i, j), photomorphogenesis (Fig. 4h), and links of kinases and carotenogenesis (Fig. 4h). Overall, the 1860 top$_{20}$ hub genes found among each of the total of 93 clusters computed by WGCNA for all three species belong to 1473 HOGs (Supplementary Data 3). Of these, 1047 (71.1%) are HOGs containing hubs across all three species (Fig. 4l), indicating their shared high connectivity across 600 million years of divergent evolution.

The clearest correlation with the apocarotenoid signals show *Pp*pink and *Me*black ($r = 0.61$, $P = 10^{-8}$ and $r = 0.41$, $P = 3 \times 10^{-4}$). *Pp*pink is enriched in light intensity including genes for the phytochrome signaling hub LONG AFTER FAR-RED 3 (LAF3)[44] and oxidoreductase activity including a superoxide dismutase hub (Supplementary Fig. 5). *Me*black shows a very complex and broad enrichment of genes associated with general transcription processes. To disentangle these genetic networks, we next used temporal clustering.

## Temporal stress co-expression and Granger causal inference of gene regulatory networks

To understand the time-course dynamics of the responses, we modeled clusters of gene expression along their time course using a Dirichlet process Gaussian process mixture model[45] (DPGP). Using DPGP, we clustered all 11,670, 9781, and 3887 genes that passed the expression cutoff in *Me, Zc*, and *Pp* into 12–16, 13–20, and 11–13 clusters (four types of clusters per condition, i.e. cold, heat, HL, HL + recovery; each set of four for the three species; Supplementary Data 2). We filtered these clusters, retaining only those that had a Gaussian probability of at least 0.7 and calculated their Jaccard distances (Fig. 5a; Supplementary Data 2). As expected, the most similar clusters occurred within a species. However, there are several similar clusters across species, for example, a cohort of genes downregulated upon heat in *Pp*C6 and *Me*C3 related to photosynthesis (Fig. 5b). *Pp* showed pronounced dynamics upon heat treatment (cf. Fig. 1d); we explored this a bit further based on temporally clustered Jaccard similar genes that show distinct temporal behaviors and found that *Pp* apparently responds with an early upregulation of genes enriched for functions in

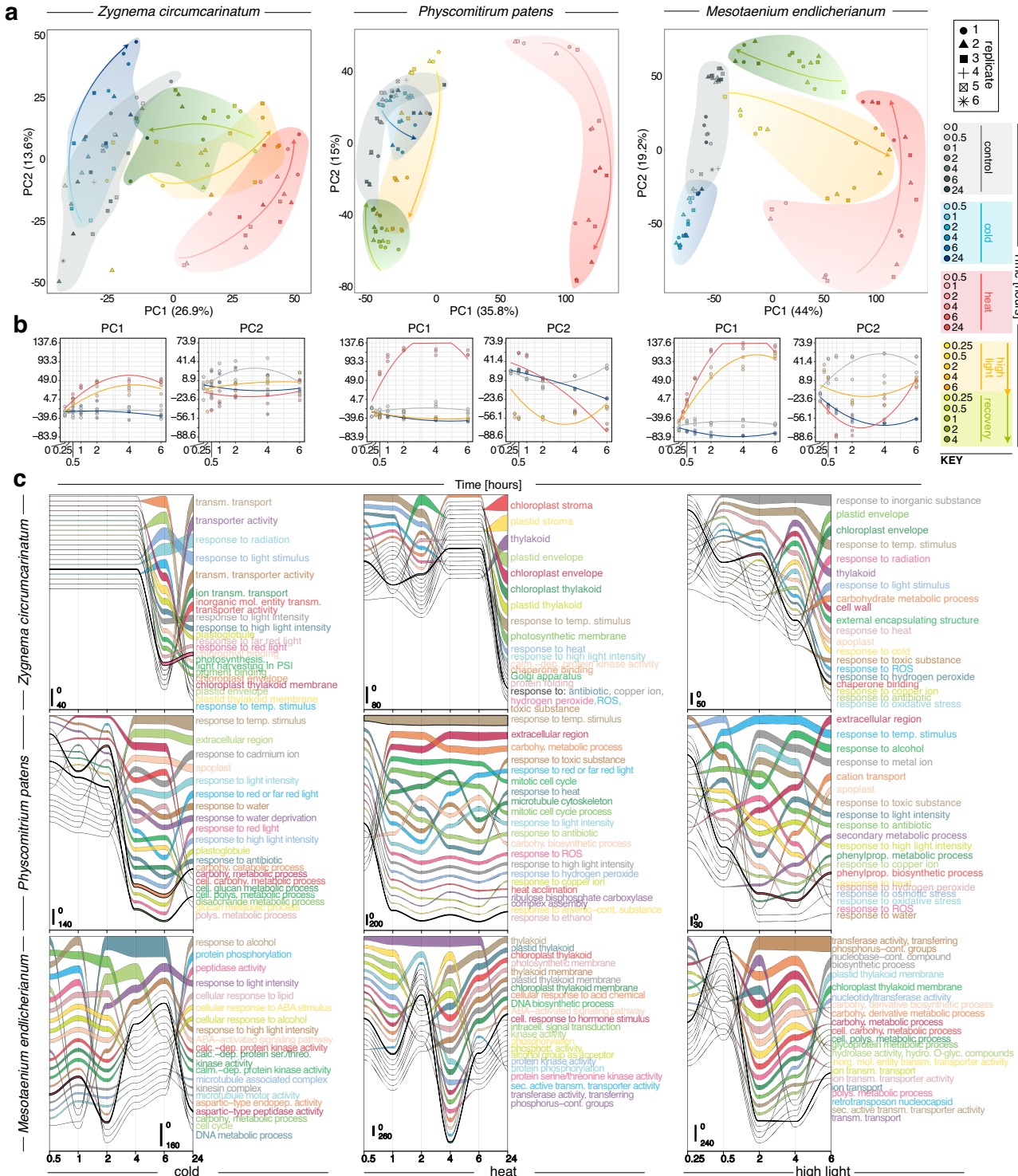

**Fig. 2 | Divergent temporal stress progression retraced by global differential gene expression analysis. a** PCA analyses for all three species with color and symbol code for the respective RNAseq samples on the right. **b** Time-dependent changes (x-axis) in the data are highlighted in plots of time versus PC1 and PC2. **c** Time-resolved alluvial diagrams of the most enriched GO terms found across at least three time points based on significantly differentially regulated genes.

Different terms are separated by color. The enrichment of a term is indicated by the width of the line (see scale in each plot). Enrichment analyses were performed by comparing the treatments cold, heat and high light stress exposure versus control. The x-axis represents the time scale in hours. Arrangement of terms along the y-axis shows which term is, at a given time point, at which rank among the most enriched.

multicellular developmental processes (*Pp*H7; links of morphogenesis, cell–cell interaction, and stress response; Supplementary Fig. 6c–e). We next turned to understand how the hubs of the co-expression clusters identified via WGCNA behave along the temporal gradient, with a focus on signaling hubs. Certain DPGP clusters are richer in

WGCNA-defined hubs; for example, hubs of *Me*yellow are mainly in a cohort of early down-regulated genes upon HL *Me*C1 and *Pp*yellow shows a similar behavior (chloroplast and photosynthesis; Supplementary Figs. 5d and 6). Early upregulated hubs include those of protein homeostasis in *Me*brown.

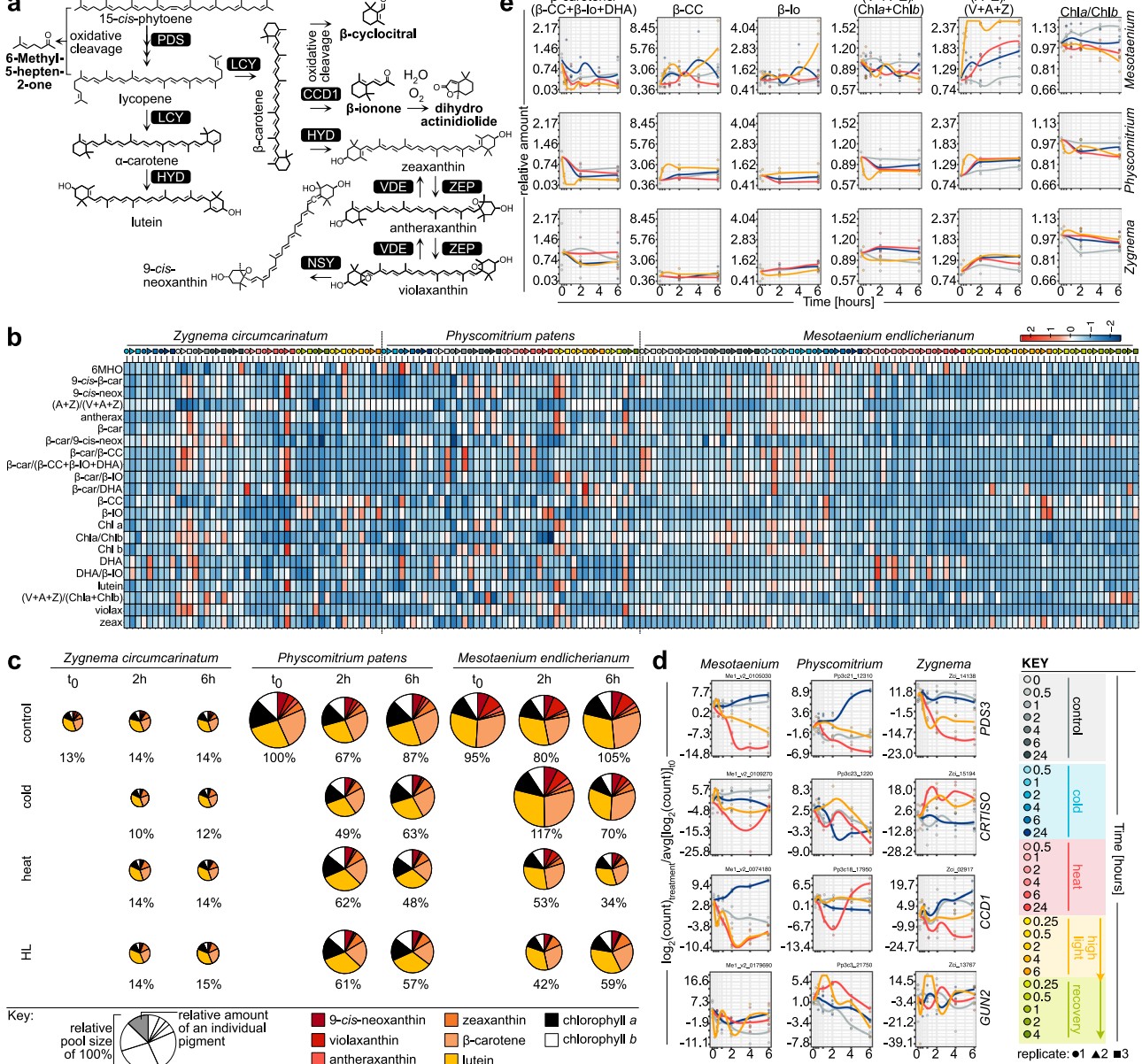

**Fig. 3 | Carotenoid and apocarotenoid dynamics upon stress. a** Simplified schematic representation of the conserved and intertwined carotenoid and apocarotenoid pathways in the three organisms of this work. **b** Heatmap of metabolite levels under the different conditions/samples; the color and symbol codes are as in Fig. 2 and can be found in the bottom right corner of the figure. **c** Pigment pool fluctuations due to stress exposure. Areas relative to size of pigment pool/contribution of individual pigment to the pool normalized on *Pp* t₀ (see key and individual values written below each pie). The total size of the pie reflects the overall size of the pigment pool; for an alternative visualization of the pigment data, see Supplementary Fig. 13. **d** Expression changes of important homologous genes likely salient to carotenoid metabolism in the first 6 h of stress exposure (only one respective homolog shown also in some cases more were present; for the expression profiles of all detected homologs and all correlations, see Supplementary Figs. 14–17). **e** Relative change of selected metabolite levels/ratios in the first 6 h of

stress exposure relative to the average value of t₀ normalized by DW of sample, recovery rate of measurement, and molecular mass of the respective metabolites. 6MHO, 6-methyl-5-hepten-2-one, 9-cis-β-car, 9-*cis*-β-Carotene, 9-*cis*-neox, 9-*cis*-Neoxanthin, (A + Z)/(V + A + Z), (Antheraxanthin + Zeaxanthin)/(Violaxanthin +Antheraxanthin+Zeaxanthin), antherax, Antheraxanthin, β-car, β-Carotene, β-car/9-*cis*-neox, β-Carotene/9-*cis*- β-Carotene, β-Car/β-CC, β-Carotene/β-Cyclocitral, β-Car/β-IO, β-Carotene/β-Ionone, β-Car/DHA, beta β-Carotene/Dihydroactinidiolide, β-Car/β-CC + β-IO + DHA, β-Carotene/(β-Cyclocitral + β-Ionone + Dihydroactinidiolide), β-CC, β-Cyclocitral, β-IO, β-Ionone, Chl *a*, Chlorophyll *a*, Chl*a*/Chl*b*, Chlorophyll *a*/Chlorophyll *b*, Chl *b*, Chlorophyll *b*, DHA, Dihydroactinidiolide, DHA/β-IO, Dihydroactinidiolide/β-Ionone, violax, Violaxanthin, V + A + Z/Chl*a* + Chl*b*, (Violaxanthin + Antheraxanthin + Zeaxanthin)/(Chlorophyll *b*), Zeax, Zeaxanthin.

Research on model land plants has established a rich framework of genetic cascades that act in information processing and are structured into a hierarchy. To understand how the temporal expression patterns of the genes are linked in the 600-million-year divergent streptophytes, we predicted gene regulatory relationships based on Granger causality using Sliding Window Inference for Network

Generation (SWING) and a Random Forrest (RF) approach (i.e., SWING-RF)[46]. Note that the predicted GRN has been reconstructed based on Granger causality which is not the same as biological regulatory interactions; it rather reflects whether gene expression changes in one gene are a significant explanatory factor in the expression change of another gene. For SWING-RF, we worked with a subset of 1897, 3694,

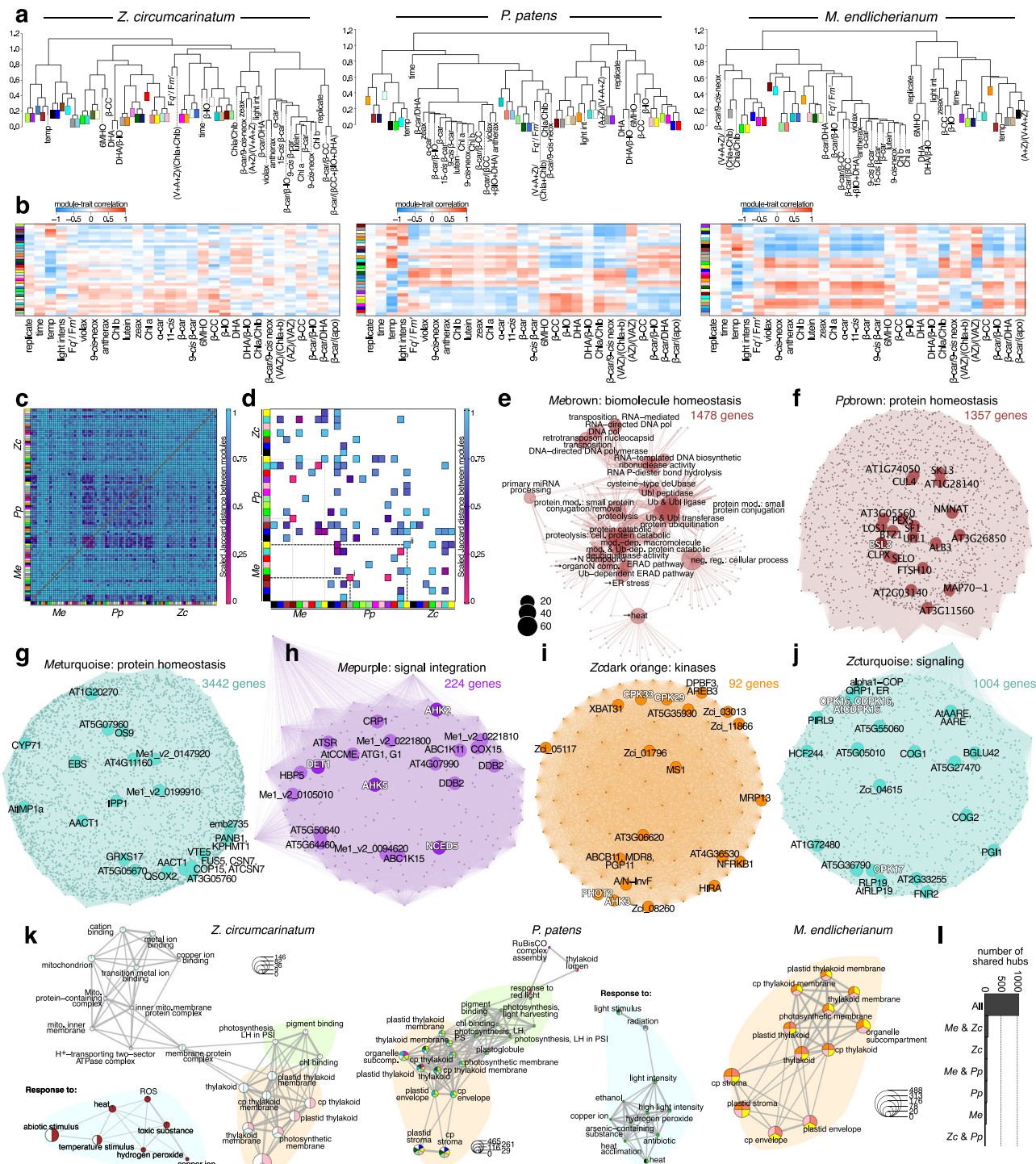

**Fig. 4 | Unsupervised gene co-expression networks recover shared programs.** WGCNA was used to compute co-expression networks. **a** Distance dendrograms of module correlation with metabolites, condition, and photophysiology. **b** Heatmaps of module trait correlation with metabolites, condition, and photophysiology corresponding to the distance trees in **a**. **c** Scaled Jaccard distances that illustrate similarity (pink) and dissimilarity (blue) between modules based on HOGs. **d** The 50 most similar modules; GO terms enriched among pairwise similar clusters are shown in Supplementary Fig. 18. **e** cnetplot of the GO terms enriched in the cluster *Me*brown. **f**–**j** Selected WGCNA clusters with the top 20 most connected genes (the hubs) annotated based on homology; next to the clusters, the total number of genes within the cluster is noted in colored font (see also Supplementary Data 2). **k** Selection of GO terms enriched in clusters defined by WGCNA in *Zygnema circumcarinatum*, *Physcomitrium patens* and *Mesotaenium endlicherianum* presented as biological theme comparison; for the whole plot, see Supplementary Fig. 4. **l** Quantification of the shared occurrence of all 1860 top20 hubs (in 1473 HOGs) in all three species based on orthogroups assignment. All networks can be explored interactively at at https://rshiny.gwdg.de/apps/streptotime.

and 1629 genes and 24 metabolite levels and ratios in *Me*, *Zc*, and *Pp*, predicting 375,273–3,358,660 non-zero interactions between genes (and metabolites), yielding putative gene regulatory networks (GRNs; Supplementary Fig. 7). We then asked the question of how the genes

that group into HOGs and show conserved behavior in DPGP clusters predict each other (Fig. 6a, b) and recovered a network of 923 conserved gene pairs (1239, 1188, and 1369 in *Me*, *Zc*, *Pp*; see Methods); in addition, we also investigated the topology of the network if we (i)

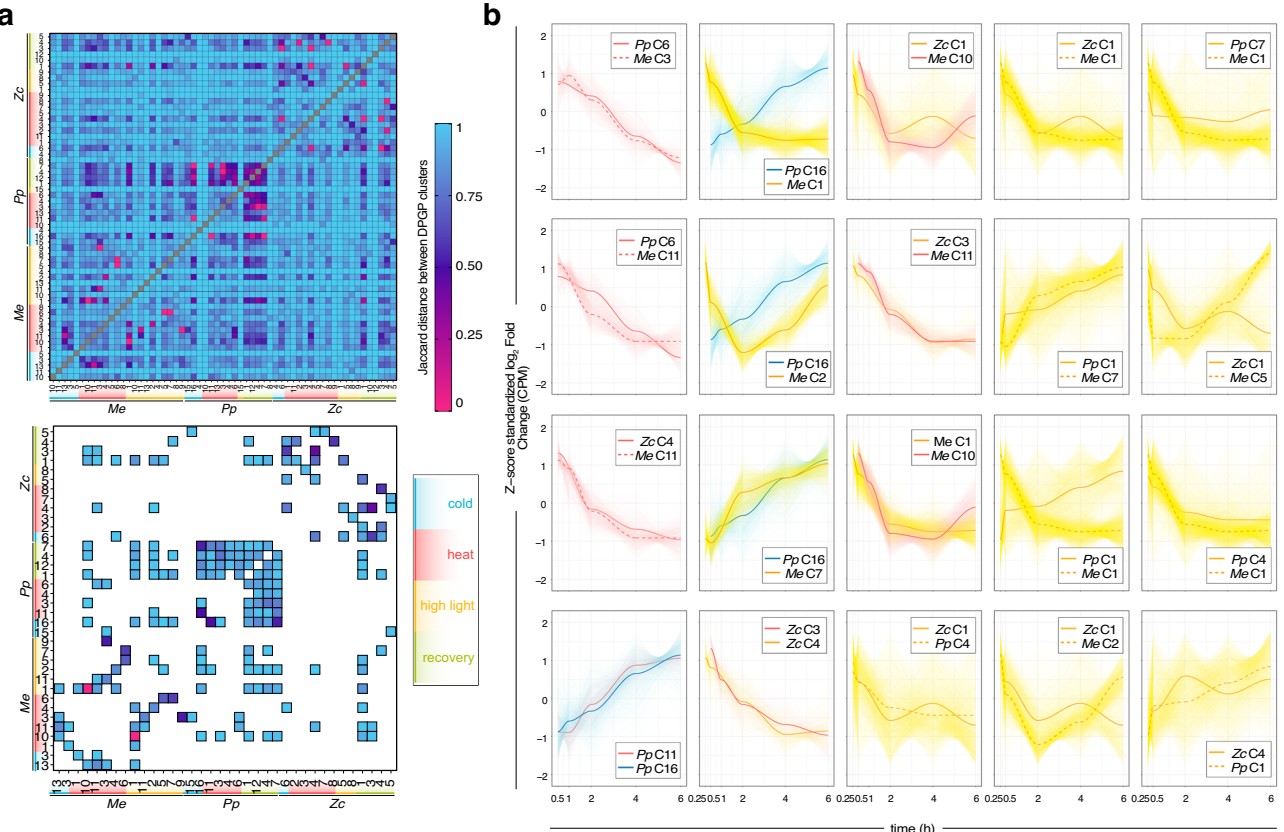

**Fig. 5 | Time-course gene expression clusters pinpoint shared responders.** Dirichlet Process Gaussian Process (DPGP) time-resolved clusters of stress samples. **a** Top: Heatmap of Jaccard distances computed for all DPGP clusters based on HOGs; bottom: zoom-in on the top 50 most similar cluster. **b** Line plots of the expression pattern of cohorts of genes recovered by DPGP clustering for 20 pairs of DPGP clusters high Jaccard similarity; the Z-score standardized $\log_2$ fold change (CPM) over the first 6 h of stress exposure is shown. The thin lines show the expression patterns of the individual genes in the clusters.

group genes simply using best BLAST hits (BBH; Supplementary Fig. 8), recovering a network of 757 conserved genes, and (ii) re-compute the network using all time points of stress treatment—also those for which no metabolite data were generated, recovering a network of 900 (Fig. 6c; Supplementary Fig. 9), and (iii) applied the same DPGP-based filtering (Fig. 6d; Supplementary Fig. 10). The topology of the predicted GRN highlighted several major points of convergence—hubs (Fig. 7a-c) signified by a high number of in degrees and characterized by high authority scores (Fig. 6a, inset); while the exact upstream and downstream relationships sometimes changed, highly connected hubs were consistently recovered throughout all analyses (cf. Fig. 6a and Supplementary Figs. 7-10; Fig. 7c) and networks followed the behavior of a scale-free network (Supplementary Fig. 11).

### Physical feedback

Mechanosensitive channels regulate ion flow in response to mechanical cues. We recover homologs coding for proteins of the OSCA family, calcium-permeable channels sensitive to hyperosmolarity[47], as the 18th most connected point of convergence (Fig. 6a; Fig. 7a, b; Supplementary Figs. 7-10). Feeding into them is a PDS1-PDS3 network (Fig. 6a), genes with highly responsive expression change under stress (Fig. 3d), underpinning a central link of carotenogenesis to all kinds of oxidative stress responses. Connected genes include those with general functions such as homologs of *CYTOSOLIC IRON-SULFUR PROTEIN ASSEMBLY 1* (*CIA1*) but also SOUL homologs (Fig. 6a). SOUL are heme-binding proteins[48] that can translocate to the chloroplast and are key for oxidative homeostasis[49].

Directly influenced by the OSCA hub is a large HOG of subtilases —the second most connected hub in the predicted GRN (Fig. 7a).

These are known growth regulators[50,51], pointing to a potential key role of peptide signaling shared by non-vascular streptophytes (see also refs. 52-54). Indeed, highly predictive for the subtilase hub genes are a CXE HOG that also contained the growth regulating gibberellin receptor gene *GIBBERELLIN INSENSITIVE DWARF1* (*GID1*)[55]—some *Me* homologs even bearing a homologous region to the unique N-terminal extension of GID1—and AHK1, which if interpreted as cytokinin-relevant, also speaks of ancient developmental programs. Indeed, upstream of CXE/GID is a HOG of *GROWTH-REGULATING FACTOR*s (*GFR*). This speaks for feedback of (i) osmo-sensing—via OSCA and AHKs[56,57] — and (ii) growth programs in which subtilases are a point of convergence. In the analysis with the extended dataset, OSCA and the subtilases remain in close proximity but another gene coding for growth-regulating proteins moved closer, a HOG of *TRICHOME BIREFRINGENCE-LIKE* (*TBL*) that act in cell wall biosynthesis[58,59] (Supplementary Figs. 9 and 10).

### Convergence at ethylene programs

Ethylene signaling stands as the sole example of a phytohormone with a clear signaling cascade reported to be functionally conserved since the common ancestor of land plants and Zygnematophyceae, although its biosynthetic pathway may not be necessarily conserved[60,61]. Exogenous ethylene treatment of the zygnemato-phyceaen alga *Spirogyra* induces growth phenotypes and the differential expression of stress- and photosynthesis-associated genes identified in land plants[62]. Crucially, *Spirogyra* homologs complement knockout lines in *Arabidopsis* ethylene signaling[63]. Ethylene thus both completes the stress and photosynthesis response cycle and represents the phytohormone with the most evident data

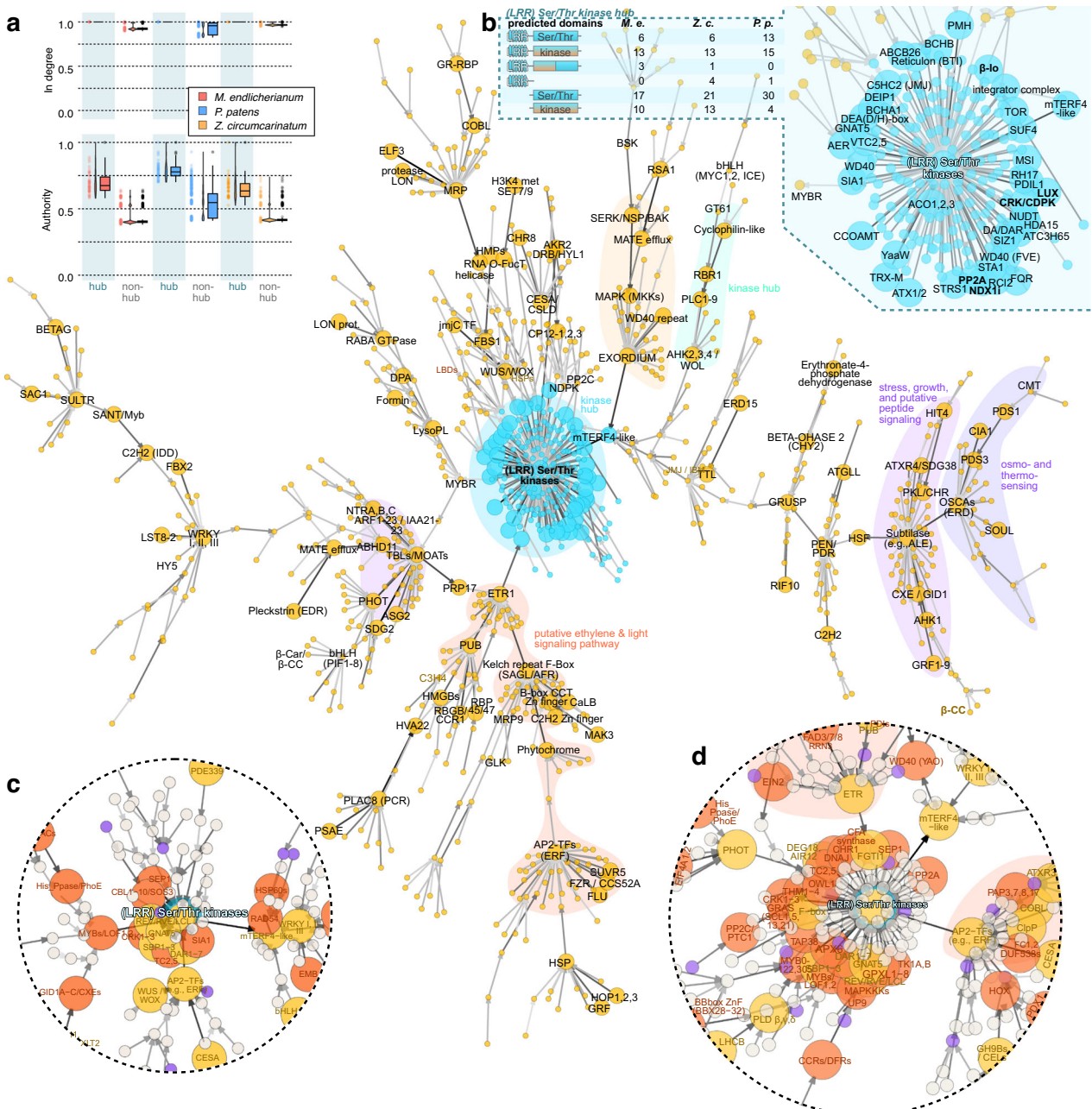

**Fig. 6 | A predicted gene regulatory network shared across 600 million years of streptophyte evolution.** SWING-RF network filtered based on conserved DPGP clusters of most predictive (enlarged circles) HOGs and metabolites conserved in all three species. All analyses via SWING and DPGP built on the log(fold changes) calculated with the package *limma*, which statistically models differential gene expression changes. All three biological replicates for each sample were used as inputs per condition. Technically, limma calculates the differences between the mean of groups as the log fold change formula. For Random forest calculations with SWING, the tree number was set to 500. **a** Predicted conserved stress responsive network including gene expression and metabolite data. Hierarchical orthogroups (HOGs) are annotated if they engage in the top 100 most predictive relationships. Note the colored putative regulatory pathways. Top left, a plot shows the parameters authority and in degree of the hub and non-hub nodes in the network; a scatter shows the data distribution. In all boxplots, horizontal lines within the boxes represent the median (Q2); the box depicts the interquartile range (IQR) from the 25th (Q1) to the 75th percentile (Q3) while the whiskers extend to

minimum or maximum values in the data determined by $Q1-1.5 \times IQR$ to $Q3+1.5 \times IQR$ (data points outside are defined as outliers). In all violin plots, we used Gaussian kernel estimation. Tails of the violins were trimmed to the range of the data and violins were scaled to the same area before trimming the tails. **b** Zoom into the central 'kinome' (*LRR*)–*Ser/Thr kinases* hub. Inset: domains predicted for the protein encoded by the genes in the *Ser/Thr kinases* hub. **c** Section of the via SWING-RF predicted gene regulatory network, re-computed with all time points for which transcriptome data were generated (without metabolite data) centered around the (also there recovered) (*LRR*)–*Ser/Thr kinases* hub; for the full network, see Supplementary Fig. 9. **d** Section of the via SWING-RF predicted gene regulatory network, re-computed with all time points for which transcriptome data were generated (without metabolite data) and filtered based on conserved DPGP clusters of most predictive (enlarged circles) HOGs centered around the (also there recovered) (*LRR*)–*Ser/Thr kinases* hub; for the full network, see Supplementary Fig. 10.

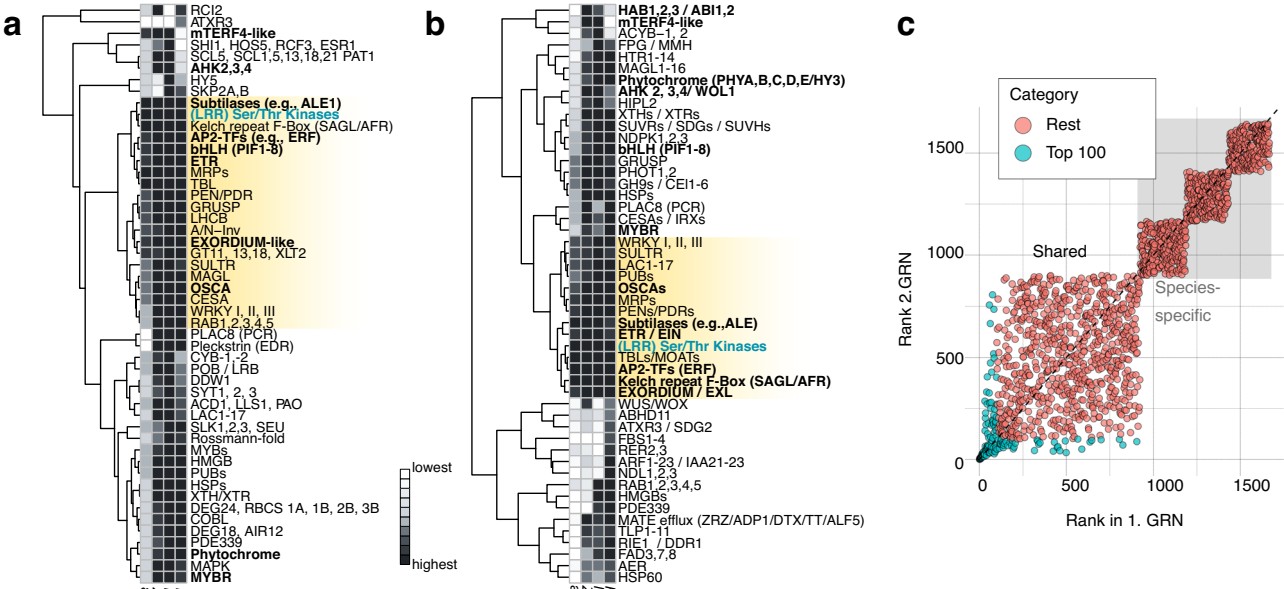

**Fig. 7 | Robust hubs in the predicted gene regulatory network. a** Ranked position of top 50 most connected nodes in the conserved network among all networks predicted by SWING (not filtered). **b** Ranked position of top 50 most connected nodes in the conserved network among all networks predicted by SWING and filtered for genes in conserved DPGP clusters. **c** Quantification of the rank of genes in the networks computed with all gene and metabolite data (1. GRN) versus the network with all time points for which transcriptome data were generated (2. GRN).

supporting its conserved functional role in stress response in both land plants and Zygnematophyceae.

Our predicted GRN recovers several signature genes from ethylene signaling. We recover *ETHYLENE RESPONSE 1* (*ETR1*) as a major hub, being the 6th most connected HOG and the topmost connected single gene in all three species (Fig. 6c, d). Both in *Arabidopsis* and *Pp*, ethylene is detected by a two-component histidine kinase receptor subfamily of ETR1 homologs[64,65]. Upstream of ETR1 in our predicted GRN are genes for meaningful regulatory cassettes including a HOG of Kelch repeat F-Box that includes homologs of the stress response and specialized metabolism regulators SMALL AND GLOSSY LEAVES (SAGL1)[66,67] and ATTENUATED FAR-RED RESPONSE (AFR)[68]. Indeed, further upstream are genes for C2H2 zinc-finger transcription factors (TFs)—TFs generally prominent in our SWING network—including SUF4 and YY1 and further well known to be among the apocarotenoid-activated TF families[28,37,69], CONSTANS-like B-box TF (that appear to encode the CCT domain) involved in light-dependent regulation[70–72], and phytochromes in line with their essential role in stress, light signaling, and several phytochrome signatures in our transcriptomes. And upstream of this is a HOG of various *APETALA2* (*AP2*) TF-coding genes forms the 13th most connected hub; this HOG includes homologs of *ETHYLENE RESPONSE FACTOR*s (*ERFs*) but have diverse functions in environmental and hormonal responses[73]. That said, in *Pp*, an ETR kinases play a crucial role in facilitating effective environmental stress responses through ethylene-mediated submergence signaling and ABA-mediated osmo-stress signaling[74]; our work adds in the conservation in the two Zygnematophyceae (Fig. 6a, d). A HOG that includes the heat stress-relevant *ARABIDOPSIS HOMOLOG OF TRITHORAX 1* and *2* (*ATX*)[75] was downstream, providing the single broker that linked it to the most connected hub in the predicted GRN: a kinase.

## Environmental and apocarotenoid input converges at hubs of the kinome

In land plants, a network of kinases regulates responses to environmental input by phosphorylating proteins like TFs and enzymes, impacting synthesis of specialized metabolites, gene activation, and more[76–78]. Several factors modulate the activity of these kinases, including protein phosphatases acting as mediators but also direct

signals if linked to a sensor, affecting response dynamics and outcomes. Such kinases form converging points in all our predicted GRNs (Figs. 6a, c, d, 8; Supplementary Figs. 7–10). They are the receivers of diverse inputs, integrating environmental input in biological programs.

As the single most connected receiver in our DPGP-filtered SWING-inferred GRN was a HOG of Ser/Thr kinases (Figs. 6a–d and 7a, b), which include several receptor-like kinases[79]. In our analyses, these receive input from the apocarotenoid ß-IO and *SUPPRESSOR OF FRIGIDA* (*SUF4*), a putative zinc-finger-containing TF gene, regulating flowering time; recently it was shown that SUF4 acts as a thermosensor, showing thermo-sensitive assembly and thus activity[80]. This warrants attention, as we recovered in the BBH-based analysis a whole putative regulatory chain of ß-IO–*SUF4*–a gene for a transmembrane protein–*MYB4R1* (Supplementary Fig. 8). Furthermore, a HOG including the circadian regulators[81] *LUX ARRHYTHMO* (*LUX*) and *BROTHER OF LUX ARRHYTHMO*, were the fifth most predictive (Fig. 7b) on the Ser/Thr kinase hub expression pattern(s), aligning with their previously noted importance in photomorphogenesis co-expression hubs in *Mesotaenium*[11]. The expression of these Ser/Thr kinase-coding genes were highly dynamic across species, in particular fast spikes early upon HL in the algae and temperature in *Pp* (Supplementary Fig. 12) reflecting the physiological responsiveness (Fig. 1c–e).

Around this hub coding for Ser/Thr kinases we recovered several genes that are known to be major hubs in regulation themselves. This included genes coding for TOR, eukaryote-wide regulators of diverse developmental and metabolic processes[82,83], which are part of the kinase network. As potential regulator of the major kinase hub, genes for PP2A B subunits appeared. This aligns with the regulatory subunit determining activation of the phosphatase complex and various subfamilies interact with different kinases[84]. Furthermore, several genes coding for proteins in calcium signaling appeared, including *CALCIUM DEPENDENT PROTEIN KINASE*s (*CPDK*s), *CDPK-RELATED KINASE*s (*CRK*s), *CALCINEURIN B-LIKE PROTEIN* (*CBL*), and even genes for putative Na⁺/H⁺ antiporters; this bolsters previous findings[11] and the idea that these represent ancient regulators[85,86].

Next to the major hub of genes for Ser/Thr kinases, other kinase HOGs included *AHK*s. AHKs are best known as cytokinin signaling

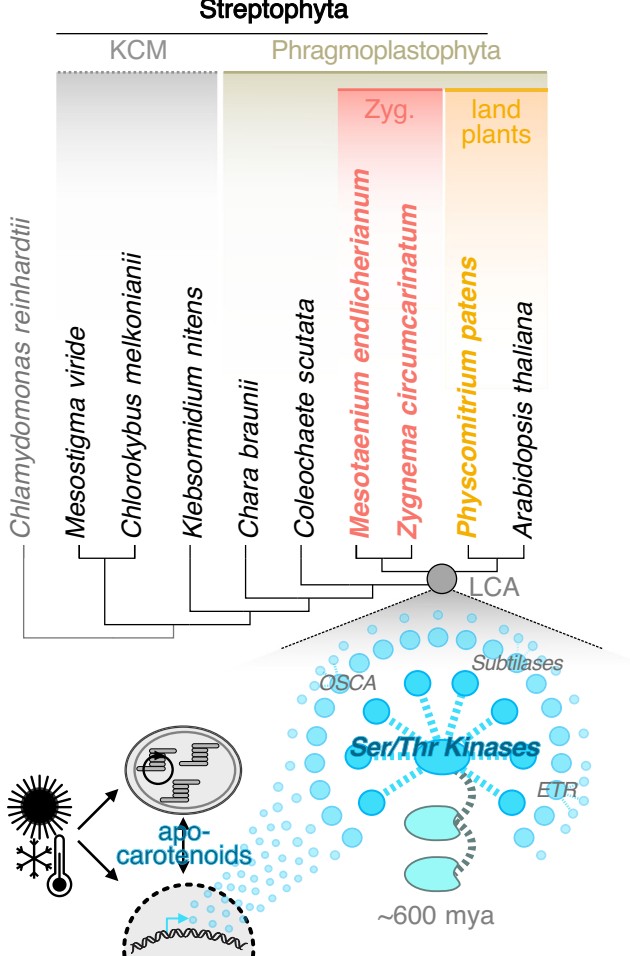

**Fig. 8 | Evolutionary assembly of environmental response cascades gleaned from transcriptome data.** In this study, the effect of light and temperature stressors on apocarotenoid levels and their concerted effect on gene expression were studied in three 600-million-year-divergent streptophytes. Our gene expression data predict an integrative network with several points of convergence, including hubs of homologous kinases, calcium/mechanosenstive channels, ethylene signalling homologs, putative peptide signalling homologs, and more.

components[87] but also include respond to diverse abiotic stressors[56,57]. These *AHK*s form another hub (being the 2nd to 10th most connected single genes) upstream of the major Ser/Thr kinase-coding gene hub. As highlighted by the previous environmental gradient analysis on *Mesotaenium*[11] alone, we find chains of kinase-coding genes (*BSK*, *SERK*, *MKK*) linked with a large group of *EXORDIUM*-like genes, which are modulators of cell expansion[88] and thus likely the downstream target that modifies growth upon environmental input. These *EXORDIUM*-likes are connected via a *mTERF4*-like HOG to the central kinome (Fig. 6). mTERF transcription termination factors harbor many important functions in regulating organellar gene expression[89,90] and mTERF4 interplays with GUN1 (see also *Zc*darkgreen, Supplementary Fig. 5) that is implicated in retrograde signaling[91].

Overall, the kinases form a major track throughout the all predicted GRNs. Many of the kinases recovered are homologous to well-known integrators of environmental cues, via intracellular signaling, to growth and acclimation programs (Fig. 8).

## On the evolutionary assembly of environmental response cascades

Interconnections of environmental input to internal programs underpin the defining capacities of land plants to adequately respond to

adverse environmental conditions[19]. In this circuitry, kinase-based cascades are some of the key wires and these cascades must have been present during the infancy of land plants. It is thus an intriguing avenue to explore the deep evolution of the respective cascades, which can be revealed through comparative analyses, such as the one performed here on three divergent streptophytes, allowing to illuminate deep origins. Research in the plant evolutionary community has yielded great insights by analyzing the evolutionary origins of important cascades through homology-guided approaches. Yet, these cascades do not always correspond to their land plant-biased "archetypal" route. Important cascade members might hence be missed, sometimes even those constituting the ancient conserved gene cores[92]. Further, since a shared signaling circuit operated in a common ancestor, divergent evolution likely brought variation to the signaling paths, despite that they might still work following a homologous biological program[92]. To find these cases, kinases can guide our search[54,92]. And that such kinase-guided searches can yield thus far unrecognized response programs that differ from the textbook is exemplified by the recent finding that auxin can act through an ancient RAF-like protein kinases mediated response system that operates in land plants and streptophyte algae[93].

Here, we use an approach that does not rely on prioritization based on homology to known key genes and pinpoint several points of predicted regulatory bundling of information (Fig. 6a inset; Fig. 8). These include several hubs of genes coding for proteins that are homologous to kinases that are known responders to recurrent themes in stress perception: sensing of the cell wall status, osmotic imbalance, recognitions of molecular patterns that can also include peptide signals and more. Co-occurring with these are genes that can underpin the generation of peptide signals (subtilase-coding genes) as well as growth regulatory that are key to cell wall homeostasis such as *EXL*. To these cascades that build on kinases we can also add the apocarotenoids that derive from pigments that any phtosynthetic organisms must have and whose production is due to physical forces and an inevitable consequence of the oxidative challenges posed by the terrestrial habitat. All mentioned aspects are recurrent themes in— and core to—the biology of the stretptophyte cell[54]. We here started piecing together parts of their molecular perception through ancient signaling cascades. These cascades constitute the ancient and shared core of how the earliest land plants responded to stress; from this core, the diversification of environmental signaling started.

## Conclusions on cue processing and the evolution of layered genetic networks in stress response

The success of the earliest land plants likely hinged on their capacity to perceive and react to environmental conditions. These environmental conditions are integrated into a network that bundles information and triggers developmental plasticity. Integrated in this web are key links between plastid and cell physiology: carotenoids and the signals they give rise to due to oxidative cleavage.

Carotenoids are among the best conserved and most ancient oxidative stress-mitigating molecules in photosynthetic organisms[24,94]. With the synthesis of the first carotenoid, non-enzymatically formed apocarotenoids were born; enzymes for controlled oxidative cleavage of the polyene backbone likely evolved later. Our data show shared stress-mitigation programs and apocarotenogenesis, driven by physicochemical conditions. What diversified were the genetic hubs and enzymes they interact with, which fine-tuned signal formation and responses likely favorable during terrestrialization. Indeed, plant terrestrialiazation was conceivably accompanied by increased oxidative force that consequently led to more apocarotenogenesis, all fostered by elevated $O_2$ levels, reduced $CO_2$ levels, temperature dynamics, and generally more abiotic stress.

Since their emergence, traits of land plants have been dependent on complex, but flexible pathways that concertedly bring about their adaptive potential. And in many cases the signature genes that

underpin these traits emerged prior to the origin of land plants[6,10,54]. Genetic networks of plants can be conceptualized in a multi-layered structure[19]. Our findings pinpoint to genetic hubs in which gene expressional changes are bundled (Fig. 6a inset), establishing links between environmental input that influences a whole layer of genes and the points in which this input consolidates. This consolidation appears to a major extent mediated by kinases (Figs. 6–8) that are known to integrate in signaling cascades that facilitate a cross-talk between different inputs, for example bundling signaling molecules, $Ca^{2+}$, and osmotic cues[57,95], and their link to developmental programs; surprisingly, this is in the analyses here gleaned from transcriptional changes. These kinases appear to span a web, serving as points of convergence between inputs and redistribute these on the level of protein phosphorylation to the respective output. We here found environmental programs that regulate the transcriptional level across 600-million-year divergent streptophytes.

## Methods

### Algae and moss cultivation

*Mesotaenium endlicherianum* SAG 12.97 and *Zygnema circumcarinatum* SAG 698-1b were obtained from the Culture Collection of Algae at Göttingen University (SAG) and grown on cellophane disks (folia®, Max Bringmann KG) as described[96] prior to stress treatment with the exception that in both cases agarized (1%) Woods Hole Medium (WHM)[97] was used. In short, fully-grown plates of stock cultures were suspended and inoculated on fresh WHM plates. *Mesotaenium endlicherianum* SAG 12.97 was grown for 8 d at $20 \pm 1\,°C$ with 80–90 µmol photons $m^{-2}\,s^{-1}$ (Niello® LED 300 W, 380–740 nm spectrum; Supplementary Fig. 1) under 16/8-h light/dark cycle. *Zygnema circumcarinatum* SAG 698-1b was grown for 48 h at $20 \pm 1\,°C$ with 20–25 µmol photons $m^{-2}\,s^{-1}$ (Niello® LED 300 W, 380–740 nm spectrum) and afterward for 11 d at $20 \pm 1\,°C$ with 80–90 µmol photons $m^{-2}\,s^{-1}$ with the same 16/8-h light/dark cycle (13 d of total growth until the start of the experiment).

*Physcomitrium patens* Gransden 2004 strain 40001 protonema was used for the experiments. Agarized (0.55%) basal minimum medium with ammonium tartrate (BCD-AT)[98] was used and inoculated with 2 mL of a suspension of the stock culture (one fully-grown plate per 20 mL sterile tab water) homogenized with an IKA® ULTRA-TURRAX®. Cultures were kept at $25 \pm 1\,°C$ during the light phase with 100 µmol photons $m^{-2}\,s^{-1}$ (growth chambers of Percival Scientific, Inc., Perry, IA, USA) and at $18 \pm 1\,°C$ during dark phase under 16/8-h light/dark cycle for 7 d prior to stress treatment or sub-cultivation.

### Stress treatments

Time-series stress experiments were performed as visualized in Fig. 1b in biological triplicates. Transcriptome and metabolite samples were harvested at the time points indicated by rectangular lines in Fig. 1b. Also, relative quantum yield was measured at these time points (for details see the photophysiological measurements section).

For the control timeline, cultures stayed for another 24 h at the cultivation conditions described above. For temperature stress treatments, control light conditions were used but the temperature was 12 °C lower (cold stress) or higher (heat stress). Due to the reduced efficiency of the LEDs at elevated temperature, the light intensity dropped by 10 µmol photons $m^{-2}\,s^{-1}$ during algae heat stress experiments (not the case for *Physcomitrium patens*).

Algae temperature stress treatments were performed in HS80 growth tents (Secret Jardin, Manage, Belgium) in temperature-controlled rooms. To ensure homogeneous temperature in the tents, cultures were put on an elevated grid and two small Clip-Fans (Garden Highpro Clip-Ventilator Ø15 cm, 5 Watt) were added. *Physcomitrium patens* temperature stress series were performed in the Percival's with altered temperature programs.

For HL treatments, cultures were transferred to a 10× light intensity regime compared to control conditions 800–900 µmol

photons $m^{-2}\,s^{-1}$ (algae) and 950–1050 µmol photons $m^{-2}\,s^{-1}$ (moss) respectively, and afterwards 4 h to control conditions for recovery. In the case of algae high light treatments, the increased light intensity also led to a rise in temperature by 12 °C. The temperature stayed constant during *Physcomitrium patens* HL treatments.

To achieve this light intensity two lamps with higher output (Niello® LED 900 W, 380–740 nm spectrum) were used in the case of the algae treatments and for *Physcomitrium patens* HL treatments, 4 additional lamps (Niello® LED 300 W, 380–740 nm spectrum) were installed in the Percival.

### Harvesting and storage of transcriptome and metabolite samples

Samples for transcriptomic analysis and metabolite profiling were harvested with a spatula at the respective timepoints and immediately frozen in liquid $N_2$ in reaction vials. Afterwards, samples were stored at −70 or −80 °C and metabolite samples additionally overlaid with argon before storage. To ensure a higher reproducibility, each transcriptome and metabolite sample was pooled from three different algae or moss plates (technical triplicate) at each timepoint. This was done in biological triplicates as described above.

### Photophysiological measurements

Fq′/Fm was determined using a MINI-PAM II (Heinz Walz GmbH) to access the photophysiological perturbations due to stress exposure at the time-points represented by rectangular lines in Fig. 1b. Only one culture of the three technical replicates (pooled for transcriptome and metabolite analysis) per timepoint and per biological replicate was measured to reduce the perturbations by the light pulse (intensity = 8 (pulse of 4000 µmol photons $m^{-2}\,s^{-1}$), frequency = 3, gain = 3) on metabolite and transcriptomic level.

### Microscopy

Microscope pictures were taken in biological triplicates. For control, cold and heat stress conditions pictures were taken after 0 h and 24 h. For the HL/recovery time series after 6 h of stress and 4 h of recovery. The microscopical setup was comprised of a Carl Zeiss Axioscope 7 RL BF/DF/C-DIC, TL LED with 10×, 20×, 40× and 100× objectives (Carl Zeiss Microscopy) connected to an Axiocam 208 color. The data was processed with the ZEN (blue edition; version 3.0) imaging system (Carl Zeiss Microscopy). For statistical analysis 6–10 (once 6 and once 8 for *Zygnema circumcarinatum*) pictures per replicate with 4–8 cells were evaluated (total of 40 cells).

### Chemicals

Carotenoid standards of 9-*cis*-neoxanthin (≥97%), violaxanthin (≥90%), lutein (≥99%), α-carotene (≥95%), β-carotene (≥93%), and lycopene (≥90%) were obtained from Sigma Aldrich Chemie GmbH and Zeaxanthin (≥90%) from ChemPur GmbH. Chlorophyll a (≥95%) and b (≥95%) were obtained from Sigma Aldrich Chemie GmbH. Apocarotenoid standards of 6-methyl-5-hepten-2-one (≥99%), β-CC (≥97%) and β-IO (≥96%) were obtained from Sigma Aldrich Chemie GmbH. Dihydroactinidiolide (≥98%) was synthesized by abcr GmbH. β-IO-D₃ (≥95%) used as the internal standard for apocarotenoid quantification was obtained from Eptes. MTBE (≥99.5%) and methanol (≥99.8%) for HPLC analysis were obtained from Fisher Scientific GmbH and acetone for extraction and standard dilutions from Carl Roth GmbH (≥99.9%). Ethylacetate (≥99.9%) for apocarotenoid standard dilutions was obtained from Sigma Aldrich Chemie GmbH and methanol (≥99.9%) for standard dilutions from Fisher Scientific GmbH. Further solvents for carotenoid standard dilutions were dichloromethane (≥99.9%) from Carl Roth GmbH and acetonitrile (≥99.9%) from Fisher Scientific GmbH. BHT (≥99.7%) was obtained from Carl Roth GmbH. The water used for HPLC analysis and extraction was purified by ultrapure water system arium pro (Sartorius).

## Carotenoid and chlorophyll extraction

Initially, the extraction protocol was inspired by Aronsson et al.[99]. Several alterations were made to optimize the protocol for organisms investigated in this study, as follows. Prior to extraction, algae samples were lyophilized (ZIRBUS technology GmbH, Bad Grund, Germany) for 18–20 h and moss samples for circa 24 h. Samples were shielded from light during this process. Next, samples were immediately frozen again in liquid $N_2$ and homogenized quickly in reaction vials placed in a metal block (blockage of light and homogenous temperature) cooled by liquid $N_2$ (cooling and replacement of oxygen) with a sharp conical spatula to prevent oxidation or degradation. Samples were overlaid with argon again and stored at −80 °C until extraction. Directly before extraction samples were frozen in liquid $N_2$ again. All extractions were performed in a temperature-controlled room at 4 °C in the dark (only indirect light with an intensity below detection limit) to prevent alterations during extraction. Extraction solvents contained 0.1w% BHT to additionally prevent oxidation. Biomass was weighted quickly in the same room and the remaining sample was directly frozen again in liquid $N_2$ until HS-SPME-GC-MS analysis. 250–700 µL solvent A (acetone:water (80:20) + 0.1w% BHT) were added immediately to the weighted sample. Extraction volume was adjusted to the available/weighted biomass to ensure similar biomass:solvent ratios. This ratio was slightly optimized for each organism (average biomass used per mL of solvent-mix: *Mesotaenium endlicherianum* 5 mgDW/mL, *Zygnema circumcarinatum* 12 mgDW/mL, *Physcomitrium patens* 5 mgDW/mL) due to different pigment concentrations in different species. After adding solvent A the mixture was directly vortexed for circa 1–2 min (2–4 min for *Zygnema circumcarinatum* due to its worse extractability). Then, the sample was centrifuged for 1 min at 4 °C at max. speed 20238 rcf (Eppendorf 5424), and the supernatant was collected in another reaction tube. The procedure was repeated with solvent B (acetone + 0.1w% BHT) but the pellet was broken by stirring it up with a pipette tip before vortexing. The supernatants were combined, vortexed for 10 s and centrifuged again at max. speed to clear the extract from insolubles. 200 µL of the extract were transferred to a brown glass analysis vial with glass insert and PTFE septum. Samples were injected (40 µL) into the HPLC within the next 5–25 min after extraction and stored on a cooled (10 °C) light shielded sample rack until analysis.

## HPLC-UV-Vis analysis of carotenoids and chlorophylls

The HPLC system Agilent 1100 series equipped with a UV-Vis-DAD detector was used for simultaneous carotenoid and chlorophyl measurements. For separation, a YMC Carotenoid C30 S- 3 µm column (250 × 4.6 mm I.D.) from YMC Europe was integrated. The solvent system used was initially inspired by Gupta et al.[100] but modified for this study, as follows. Eluents were degassed in an ultrasonic sonic bath (≥15 min) before connection with the HPLC system. A ternary gradient of eluent A (methanol:water, 98:2, v/v), eluent B (methanol:water, 95:2, v/v), and eluent C (MTBE) was applied as follows: 0 min A:C (80:20, flow rate 1.4 mL/min), 0–2.00 min gradient to A:C (70:30, flow rate 1.4 mL/min), 2.01 min B:C (70:30, flow rate 1.0 mL/min), 2.01–18.00 min gradient to B:C (0:100, flow rate 1.0 mL/min), 18:00–19.00 min gradient to A:C (80:20, flow rate 1.0 mL/min), 19.00–25.00 flow rate gradient from 1.0 mL/min to 1.4 mL/min A:C (80:20), hold for 1 min on this condition (total run time 26 min). The oven temperature was 20 °C.

Identities of the detected molecules were determined by absorption spectra, retention time, and comparison of both with authentic commercial standards (if available). In case of antheraxanthin[100] and respective *cis/trans*-isomers of carotenoids[100] identities of the detected molecules were determined by absorption spectra by comparison with literature data. Violaxanthin, lutein, α-carotene, β-carotene, and zeaxanthin were quantified at 451 nm based on calibration curves of the respective standards (Supplementary Methods Figs. 2–4, and Supplementary Methods Table 1). 9-*cis*-β-carotene was quantified based on β-carotene calibration due to similar spectral properties[101]. The analytical standard of 9-*cis*-neoxanthin was an almost equal mixture of isomers, so it was quantified based on violaxanthin calibration respecting differences in extinction coefficients instead[102] (Supplementary Methods Table 2). Antheraxanthin (Supplementary Methods Table 2) was quantified based on zeaxanthin calibration respecting differences in extinction coefficients[102]. Chlorophyll a and b were quantified at 660 nm based on calibration curves of the respective standards (Supplementary Methods Figs. 2–4, and Supplementary Methods Table 1).

## HS-SPME-GC-MS analysis of volatile apocarotenoids

The method established here was initially based on Rivers et al.[103]. The system used for volatile apocarotenoid measurements was comprised of the following technical compartments: GC/MSD instrument (Agilent Technologies 7890B) coupled to a 5977B MSD quadrupole, PAL3 Auto sampler system with Robotic Tool Change (RTC 120), polydimethylsiloxane/divinylbenzene/carboxen (50/30 µM DVB/CAR on PDMS) adsorbent SPME fiber from Supelco, HP-5MS UI column (30 m × 0.25 mm; 0.25 µm coating thickness; Agilent).

The temperature program for elution was the following: Inlet 250 °C, 60 °C 2 min hold, ramp 5 °C/min till 185 °C, ramp 25 °C/min till 320 °C followed by a 2 min hold, in total 34.4 min. Post run was set to 60 °C and Aux heater (MSD Transfer Line) to 280 °C. Helium gas flow rate was set to 1 mL/min.

Metabolite adsorption and desorption program: 30 min preconditioning at 270 °C, 5 min sample equilibration at 70 °C with agitation, 40 min sample adsorption on fiber at 70 °C with agitation at a penetration depth of 40 mm followed by 35 min of sample desorption in the inlet at 250 °C.

The electron impact ionization energy (EI) was set to 70 eV and the ion source temperature was 230 °C.

Identities of the apocarotenoids were determined by fragmentation patterns and retention times by comparison with authentic analytical apocarotenoid standards in each species. For that purpose, total ion chromatograms (TIC) were recorded (m/z 40–300). All apocarotenoids were confirmed in all organisms by this measure except 6-methyl-5-hepten-2-one which was only detectable in selective ion monitoring (SIM) in *Zc* and *Pp*. For quantification SIM was used and recovery rates were accessed by co-injection and co-calibration (of each standard) with β-IO-$D_3$ (Supplementary Methods Fig. 5); for respective chromatograms and fragmentation patterns, see Supplementary Methods Figs. 6–12. Ions for quantification of the respective molecules: 6-methyl-5-hepten-2-one (m/Z: 108), β-cyclocitral (m/Z: 137) and β-IO (m/Z: 177), β-IO-$D_3$ (m/Z: 179 + 180), and dihydroactinidiolide (m/Z: 111).

The remaining refrozen samples used beforehand for carotenoid quantification (see above carotenoid and chlorophyll extraction) were weighted swiftly into brown glass HS-SPME-vials (20 mL, magnetic caps with silicone/PTFE septa) in a temperature-controlled room at 4 °C in the dark (only indirect light with an intensity below detection limit) to prevent alterations by carotenoid degradation and co-injected with 4 µL β-IO-$D_3$ standard solution (same amount and concentration as for co-calibration). To reduce contamination by production residues and other volatiles in the HS-SPME-vials, they were precleaned as follows: Vials and caps were washed two times with methanol (≥99.9%) and dried overnight at 80 °C to emit remaining volatiles. After cooling down the vials were closed with the caps and stored until analysis.

## RNA extraction

The RNA extraction was based on Dadras et al.[11] and slightly optimized for each organism. In general, the protocol described by the vendor of Spectrum Plant Total RNA Kit (Sigma) was used. Changes are described in the following. Frozen *Me* SAG 12.97 samples were put on ice, mixed

immediately with 500 μL lysis buffer containing 2-mercaptoethanol (10 μL/mL lysis buffer), vortexed briefly and transferred for one minute to an ultrasonic bath for optimal cell penetration followed by 5 min of heat shock (56 °C). Protocol B (increased binding solution (750 μL)) was continued as described by the vendor. For *Pp* extraction, the same protocol was used as for *Me* but with 1 mL lysis buffer and four minutes of ultrasonic bath. Frozen *Zc* SAG 698-1b samples were lyophilized freshly before extraction for 18–20 h (ZIRBUS technology GmbH, Bad Grund, Germany), frozen again in liquid $N_2$, put on ice, mixed immediately with 1 mL lysis buffer, vortexed briefly and transferred for four minutes to an ultrasonic bath for optimal cell penetration followed by 5 min of heat shock (56 °C). Protocol B was continued as described by the vendor.

## Functional annotation

In order to assign functional information to the sequences we employed a comprehensive set of tools that we bundled, together with all the other tools and code used, in a workflow deposited on Zenodo[104]. These tools included InterProScan[105] (v5.64-96.0 and -pa -goterms flags), eggNOG-mapper[106–108] (v2.1.12 and -m diamond--dmnd_iterate yes--dbmem --cpu 0 --evalue 1e-10 --sensmode ultra-sensitive --tax_scope 33090 --dmnd_db eggnog_proteins_default_viridiplantae.dmnd flags), BLAST[109] (v2.15.0) against protein files of *A. thaliana*[110] and genome scale gene family analysis using Orthofinder[111–114] (v2.5.5). *Me*, *Zc* SAG698-1b and *Pp* had 74.5, 93.9, and 78.1% of their genes in HOGs and they have 8.2, 2.2, 12% or their genes in species-specific orthogroups, respectively. We first ran Orthofinder with these settings: -S diamond -M msa -A mafft -T fasttree -t 200 -a 6 -y. Building on this, we redid the analysis by providing a user-defined rooted species tree to increase the accuracy of the inference and this tree includes the following species: *Anthoceros agrestis* oxford[115], *Azolla filiculoides*[116], *A. thaliana*[110], *Brachypodium distachyon*[117], *Chara braunii*[6], *Chlorokybus melkonianii*[9,118], *Chlamydomonas reinhardtii*[119], *Closterium sp.* NIES-67[120], *Klebsormidium nitens*[5], *Mesotaenium endlicherianum*[7,11], *Marchantia polymorpha*[121], *Mesostigma viride*[9], *Ostreococcus lucimarinus*[122], *Oryza sativa*[123], *Prasinoderma coloniale*[124], *Penium margaritaceum*[8], *Physcomitrium patens*[39], *Solanum lycopersicum*[125], *Selaginella moellendorffii*[126], *Spirogloea muscicola*[7], *Zygnema circumcarinatum*[10] SAG 698-1a, *Z. circumcarinatum*[10] SAG 698-1b, *Z. circumcarinatum*[10] UTEX 1559, *Z. circumcarinatum*[10] UTEX 1560, and *Zea mays*[127]. To assign GO terms to each gene, we combined the functional annotation of InterProScan and eggNOG-mapper into a table for each species using the ontologyIndex package (v2.11)[128]. We also used Tapscan (v3)[129] to identify transcription factors for each species.

## Quality control and gene expression quantification and exploratory data analysis

We used the pipeline of Dadras et al.[11] built using Snakemake (v7.7.0)[130] and available on GitHub (https://github.com/dadrasarmin/rnaseq_quantification_kallisto_pipeline). Briefly, we used FastQC (v0.12.1)[131] and MultiQC (v1.16)[132] to perform quality control, Trimmomatic (v0.39)[133] to perform trimming and filtering, and Kallisto (v0.48.0)[134] to quantify gene expressions (see also Supplementary Methods).

We used R (v4.3.2)[135] and tidyverse (v2.0.0)[136] for data analysis and visualization. We used tximport (v1.30.0)[137] to import and summarize count tables at gene-level into R and normalized count tables for both sequencing depth and gene length using the following settings: "countsFromAbundance = "lengthScaledTPM", txOut = F". We used edgeR (v4.0.6)[138] to keep only genes with expression levels higher than 10 counts-per-million (CPM) in at least 3 samples. Based on the experimental design of this study, we chose to perform global normalization (quantile normalization) to remove technical unwanted variations in our dataset[139]. We used qsmooth (v1.18.0)[140] with treatments as group_factor to perform the normalization.

## Differential gene expression analysis

We used limma (v3.58.1)[141] to model gene expression changes under each treatment samples compared to the same time point in the control condition using lmFit, contrasts.fit, eBayes, and decideTests functions. We picked |log2(fold change)| ≥1 as well as Benjamini–Hochberg method for *p*-value adjustment and a threshold of 0.05 to determine differentially expressed genes (DEGs). We used GO-gene tables that we prepared in the functional annotation step to perform Over-Representation Analysis (ORA) using clusterProfiler (v4.10.0)[142]. In this section, we only focused on "Biological Process" domain of GO terms, using only expressed genes in our dataset as background, adjusted *p*. value cut off ≤0.05 and q. value cut off ≤0.05 for enrichment analysis. To visualize the general pattern of GO term enrichment overtime under each treatment, we used alluvial (v0.2.0)[143] and picked top 10 GO terms that are enriched in as many as possible time points and sorted them on Y-axis based on the enriched gene count of the GO term. The thickness of each stratum is visualized based on the number of enriched genes in each GO term.

## Co-expression network analysis

It is well known that gene co-expression methods, each with its own strengths and weaknesses, can lead the different final networks[144]. In this study, we used two methods from different classes of co-expression network analysis. First, we use Weighted Gene Co-expression Network Analysis (WGCNA v1.72.5)[145] to infer one network from all treatments and time points per species. In this method, correlation measures are used to calculate an adjacency matrix using a beta and a network type. Next, the topological overlap matrix is calculated based on the adjacency matrix, then a distance matrix will be calculated and using hierarchical clustering genes will be divided into various modules. Finally, modules that are very similar based on their Eigenvalues will be merged. We followed the authors' recommendations for the parameters for this last step. In summary, we screened soft-thresholding powers from 1 to 50 for each species and picked a soft threshold based on mean connectivity (around 50), median connectivity (around 20), and signed $R^2$ of Scale free topology model fit (above 0.8). We picked 20, 20, 14 as soft thresholds for *Me*, *Pp*, and *Zc*. We built our networks using the following settings: Merging threshold = 0.20, correlation method = biweight midcorrelation, network type = signed, TOMType = signed, minimum module size = 30, and maximum percentile of outliers = 0.05. We calculated Pearson's correlation coefficient and gene significance based on module's Eigengene values and various physiological measurements and metabolite concentration changes. We also calculated inter- and intra-modular connectivity for each module and picked the top 20 highly connected genes as the hubs of that module. For each module, we performed GO enrichment analysis similar to the differential gene expression analysis mentioned above. Biological theme comparison plots were made using clusterProfiler to discover patterns of GO enrichment among different modules. We used igraph (v1.6.0)[146] to visualize co-expression network for each module and annotate the hubs. We annotated hubs based on the BLAST results described in the functional annotation above in this order: (a) gene symbol> (b) *A. thaliana* best hit gene ID> (c) species gene ID. All results obtained via WGCNA have been uploaded on Zenodo.

The second method is the Dirichlet Process Gaussian Process mixture model (DPGP), a non-parametric model-based method that is designed to perform gene co-expression analysis for time series datasets. It solves the problem of the number of clusters using a Dirichlet process and then model the dependencies in gene expression profiles between time points using a Gaussian model[45]. We used fold change values that has a significant adjusted *p*-value ≤ 0.05 compared to the same time point in control as the input of the software. Due to assumptions of this method, we had to make one network per species (*Me*, *Zc*, *Pp*) and per treatment (cold, heat, HL); nine networks in total. We visualized expression profiles and performed GO enrichment

analysis as mentioned above. The authors of DPGP suggested that this tool can be used to look for tightly regulated genes by filtering for gene assignments to clusters with a specific threshold in the final probability. We picked probability ≥0.7 as our threshold as suggest by the DPGP authors and compared inter- and intra-species similarities between clusters using Jaccard distance.

We picked a collection of the most similar filtered gene clusters based on Jaccard distances to investigate further. We normalized the data between 0 to 1 to visualize it as a heatmap. Also, we put a minimum cap of 0.9 Jaccard distance for both clustering methods since the heatmap was not informative due to presence of few outliers in pairwise combinations (very close clusters).

### Gene regulatory network (GRN) inference

There are various methods to calculate GRN based on time series transcriptomics but the balance between run time and accuracy makes it hard to pick a gold standard among all methods. Here, we used Sliding Window Inference for Network Generation (SWING)[46] to account for our temporal information which is one of the best method for this purpose according to independent benchmarkings[147]. SWING uses a multivariate Granger causality to infer network topology from time series data. We combined the transcriptomics data with metabolite concentrations as inputs and used the Random Forest (RF, i.e. SWING-RF) method to infer the network which has the best performance compared to LASSO and PLSR in the benchmarking[147]. The parameters that should be defined to infer the network were decided based on the best practice that was suggested by the authors of SWING as follows; For $Me$, we had more metabolite data and we picked: minimum lag = 0, maximum lag = 1, fixed-length of sliding window = 4 and number of trees = 500. For Pp and Zc, we picked these parameters: minimum lag = 0, maximum lag = 1, fixed-length of sliding window = 2 and number of tress = 500. We performed Z-score transformation on the input datasets. To integrate scores from many windows and delays into a single score (regulator-regulated pairs), we utilized this package's mean-mean aggregation approach. Confidence values from windowed subsets are combined into a single network by calculating the mean rank of the edge at each delay k, followed by the average rank of the edge over all delays.

The outcome of this method is a ranked list of all possible pairs ordered from the most to the less confident one. We first filtered out pairs with 0 support, extracted the top 0.1%, and visualized the result via igraph. Since it was still a very big network, then investigated the network with more filtering. Basically we created file list based on keywords downloaded from TAIR. (a) Cold consists of "cold acclimation", "response to cold", and "cellular response to cold" (b) Heat consists of "response to heat", "heat acclimation", "cellular response to heat", and "cellular heat acclimation" (c) HL consists of "response to high light intensity", and "cellular response to high light intensity" (d) Oxidative consists of "response to oxidative stress", "cellular response to oxidative stress", "cellular response to reactive oxygen species", "response to photooxidative stress", and "regulation of response to oxidative stress" (e) "Carotenoid metabolic process" (f) "Apocarotenoid metabolic process". We then used the BLAST results mentioned above to find possible homologs of these genes in our species of interest. We used these gene sets as well as metabolite list and TF list extracted using TapScan to look for top 0.1% edges of GRN for each of these subsets. We used igraph to visualize the data and annotate the top 100 nodes in the edge list as explained in the co-expression network section.

Based on the input, theoretically possible network could consist of 3,596,712, 13,641,942, and 2,652,012 edges for the networks of $Me$, $Pp$, and $Zc$, respectively. However, we recovered networks with 3,557,677 (98.9147%), 3,910,293 (28.66376%), and 1,415,350 (53.36891%) non-zero predictive pairs of genes ($Me$, $Pp$, and $Zc$, respectively). To calculate the conserved network, we mapped gene IDs to HOGs to find a common concept across species. The number of edges (the predictive pairs) are 1,465,716, 1,246,870, 856,719 for $Me$,

$Pp$, and $Zc$, respectively. We had 5552 HOGs that are shared between $Zc$, $Me$, and $Pp$. Therefore, in theory the maximum number of predictions (pairs of genes) is 30,819,152. The network that was conserved among $Zc$, $Pp$, and $Me$ had 383,984 (1.245927%) unique pairs. Finally, we added an extra layer of filtering by just looking at the genes that were among conserved DPGP clusters across species with probability bigger than 0.7. Here, we are looking at genes that are both tightly clustered across species (DPGP) and have conservation of clustering across species (Orthofinder and Jaccard distance). Using these as filters on top of the SWING method, significantly reduce the number of genes, but it is the set of genes that are show tight regulation at expression level as well as homology besides the Granger causality. Here, we had 923 HOGs and metabolites that were present in conserved DPGP clusters. Therefore, in theory, we could have maximum 851,006 edges in the conserved network. We had 923 edges (0.1084599%). In summary, the conserved gene pairs (923) filtered based on similar clusters of co-expression networks (DPGP) and homology (Orthofinder) are a very small portion of the potential/theoretical network—obtained after applying several filters for conservation and robustness.

### Statistics & reproducibility

Sample sizes (for RNA-Seq as well as the other experiments) were chosen based on the best practices of the field (at least 3 biological replicates per condition). Each analysis involved millions of pooled cells, all can be assumed to behave similarly (as they were vegetative cells from the same starting culture). For each species, at minimum 90 samples for RNAseq and physiological data points were analyzed. Sequencing was then performed to a depth that was chosen based on approaching saturation level (based on obtaining differential expression patterns among the given number of genes). All data were used except for an additional set of samples that turned out to be irrelevant for the study (constant light, only acquired for $Me$). The experiments were not randomized, yet all experiments are based on a random selection of millions of cells from a minimum of three pooled cultures on agar plate per replicate. The start culture was one homogenous culture that was equally distributed to inoculate the plates. A random selection of millions of cells thus ended up in one plate that was exposed to a certain condition. Prior to start, all plates were thus equal. The Investigators were not blinded to allocation during experiments and outcome assessment. Blinding was not relavant for this study. It is irrelevant for the bioinformatics because we worked with all versus all comparisons, unsupervised methods and all pipelines are fully transparent. All cell-based evaluation is quantifyable and unambigous. Further, the information is fully provided and re-evaluable.

### Reporting summary

Further information on research design is available in the Nature Portfolio Reporting Summary linked to this article.

## Data availability

All RNAseq reads have been uploaded to NCBI SRA and can be accessed under Bioproject PRJNA895341 (*Mesotaenium*) and PRJNA939006 (*Zygnema* and *Physcomitrium*) and SRA accessions from SRR22077315 to SRR22077409 (*Mesotaenium*) and from SRR23625966 to SRR23626145 (*Zygnema* and *Physcomitrium*). Raw metabolite profiling data are available on Zenodo: https://doi.org/10.5281/zenodo.10805605. Details on WGCNA are available on Zenodo: https://doi.org/10.5281/zenodo.14234484. Data can be interactively explored at https://rshiny.gwdg.de/apps/streptotime/ Source data are provided with this paper.

## Code availability

Code is available under: https://gitlab.gwdg.de/armin.dadras/time-resolved-oxidative-signal-convergence-across-the-algae-embryophyte-divide. Code is also available on Zenodo: https://doi.org/10.5281/zenodo.14710622.

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

## Acknowledgements
We thank René Heise, Sabine Freitag, Malte Bürsing, and Tarek Morsi for excellent technical support and Greta Lisbach for help during experimentation. J.d.V. thanks the European Research Council for funding under the European Union's Horizon 2020 research and innovation programme (Grant Agreement No. 852725; ERC-StG "TerreStriAL"). J.d.V. and I.F. are grateful for support through the German Research Foundation (DFG): on the grant SHOAL (514060973; VR132/11-1) and within the framework of the Priority Programme "MAdLand – Molecular Adaptation to Land: Plant Evolution to Change" (SPP 2237; 440231723 VR 132/4-1; VR 132/4-2; FE 446/14-1), in which T.R. is a PhD student and A.D., J.M.R.F.-J, and I.I. partake as associate members. A.D. is grateful for being supported through the International Max Planck Research School (IMPRS) for Genome Science. J.M.R.F.-J. and T.R. gratefully acknowledge support by the Ph.D. program "Microbiology and Biochemistry" within the framework of the "Göttingen Graduate Center for Neurosciences, Biophysics, and Molecular Biosciences" (GGNB) at the University of Goettingen. I.I. was supported by MICIU/AEI/10.13039/501100011033, ESF+ and ERDF/EU (Grants RyC2022-038245-I and PID2023-152168NB-I00). Illustrations of algae and *Physcomitrium* in Fig. 1b by Debbie Maizels, Zoobotanica Scientific Illustration. We are grateful to T. Friedl for supporting us with access to the facilities of the Department of Experimental Phycology and SAG Culture Collection of Algae. We thank the *Gesellschaft für wissenschaftliche Datenverarbeitung mbH Göttingen* (GWDG) for providing excellent computational infrastructure.

## Author contributions
J.d.V. conceived the project; J.d.V. coordinated the project with I.F.; J.d.V. and T.P.R. designed the experiments; T.P.R., T.D., S.P., N.H., C.H., S.d.V., I.I., J.M.R.F.-J and T.P. performed experimental work. A.D. and J.d.V. designed the computational analysis. A.D. carried out computational analysis. T.P.R. and C.H. performed analytics. C.H. and I.F. supervised the analytics. R.P. and S.A.R. predicted streptophyte transcription factors. J.d.V., A.D., J.M.R.F.-J, T.P.R., and S.d.V. designed figures. J.d.V., A.D., and T.P.R. contributed to writing the manuscript. J.d.V. organized and wrote the final manuscript. All authors commented, discussed, and provided input on the final manuscript.

## Funding

## Competing interests
The authors declare no competing interests.
