## [Transparent Peer Review file · Nature Communications]

Time-resolved oxidative signal convergence across the algae–embryophyte divide

Corresponding Author: Professor Jan de Vries

Version 1:

Reviewer comments:

Reviewer #1

(Remarks to the Author)

In this study, Rieseberg et al. generated extensive transcriptomic and metabolomic datasets across three streptophytes that diverged 600 million years ago to study stress response kinetics. Through network analysis, the authors discovered that major kinases and ethylene signaling components predate ancient convergent signaling pathways. I would like to express my appreciation for their efforts in providing many valuable and high-quality resources for the community. Additionally, both the analytical pipeline and the bioinformatic analysis were well-designed. Overall, this study is novel and has significant merits for various research areas. However, the manuscript could benefit from some reorganization of the text and data analysis, as well as data visualization.

Major points:

The descriptions in the results section are overly descriptive. The authors' claims would be stronger if they presented their data in a more quantitative manner.

1. Line 76-77: I didn't find the data correlating time-resolved transcriptome and metabolomic measurements. The stated correlation between traits (metabolite contents) and gene expression (modules) does not clearly reflect the kinetics of time-resolved data.
2. Fig 1c: The kinetics in Pp appear to differ significantly from the other two algae and are particularly sensitive to temperature shifts, especially during heat stress. Can the authors provide an explanation for this? Also, what are the cell numbers after 24 hours? Is the recovery of Fq'/Fm' in Pp at 24 hours related to this?
3. Line 115: Given the recent publication of a high-quality whole genome in the three species, can the authors explain the lower mapping rate in Zc?
4. Line 118: I do not see the support for the claim that "HL showed swift changes in all three species". The separation of PC1 for HL in Pp is not evident.
5. Extended Data Fig 2: Pp seems to recover faster. Can the authors comment on this observation? It also appears that the kinetics differ significantly among the three species according to the PCA analysis. Additionally, the GO analysis suggests that different terms are enriched in the three species. Please clarify in the text.
6. Line 130, Extended Data Fig 3: I can't locate the terms mentioned in the figure. It would be helpful if the authors could highlight these terms directly in the figure.
7. Fig. 2b: Does the GO enrichment pattern remain consistent when only orthologous genes are used for analysis? Are the observed differences primarily due to orthologous or non-orthologous genes? Furthermore, the authors combine three different types of GO terms (BP, MF, and CC) but only discuss BP terms in the main text. The clarity of the figure could be improved by focusing solely on BP functions.
8. Line 130: Could the authors provide additional details about the enzymes in each species? Specifically, information on how many orthologs exist in each species, along with their locus ID numbers and gene names, would be useful.

9. Fig 3b: The authors use the term "flux" in the legend; however, they only measure the abundance changes at different time points of the treatments. I'm unsure if "flux" is the correct term to use in this context.
10. Line 140, Fig 3d: The term "relevant gene" needs a clearer definition in the main text. Additionally, the legend states "one respective homolog shown"; how do the authors select which homolog to display in the figure? What about the other homologous genes in each species? Also, how are the genes in Fig 3d related to those in Fig 3a? Are they all key enzymes shown in Fig 3a?
11. Lines 146-147: It is unclear which enzymes or pathways are favored by which stress conditions based on these figures. Additionally, the gene expression trends in Fig 3d for high light (HL) and heat stress (HS) appear quite similar. Could the authors clarify this point?
12. Line 160: The results here indicate no correlation between metabolites and gene expression, whereas Line 77 in the introduction mentions a correlation. The authors should provide clearer evidence to support the correlation between these data.
13. Line 179: It's unclear from Fig. 4d; perhaps the authors could highlight the relevant parts of the figure. Additionally, marking the genes within the network could enhance clarity.
14. Fig. 4: How many genes are in each module? Do modules with similar functions from different species contain comparable numbers of genes or homologous gene group (HOG) numbers? Are the similar GO terms derived from modules based on the analysis of orthologs or all genes?
15. How do the authors select hub genes for display in the networks?
16. Line 187: Zc appears to be markedly different. Can the authors provide an explanation for this?
17. Line 206: Why are there different numbers of clusters in each species? Which number of clusters did the authors use in the formal analysis?
18. Fig. 5 & Fig. 6: The authors predict several novel genes involved in the convergent signaling network and infer numerous potential regulations among these interactions. However, the entire section (Lines 202-324) reads more like a discussion rather than a presentation of results.
19. Discussion Section: In the time-course Gene Regulatory Network (GRN) analysis, the authors argue that kinases form a network, acting as convergence points between inputs and outputs. However, the kinetics of these enzymes appear to vary significantly among the three species, and the dynamics of metabolites also differ. It would enhance clarity if the authors could link Fig. 5 to Fig. 3 and highlight the critical enzyme versus metabolite changes in Fig. 6.

Minor points:

1. There is no Fig. 2c mentioned in the manuscript.
2. Line 152: Fig. 3e does not depict gene expression.
3. What does the grey line (or grey area) represent in Fig. 5c?
4. There is no Fig. 5d in the manuscript.
5. Fig. 6: What is the rationale behind selecting these particular hubs for display?
6. The figure legends could be more precise, particularly the descriptions in the Gene Regulatory Network (GRN) sections, such as those in Lines 734 and 828.
7. Lines 831 & 833: It appears that some words may be missing.

(Remarks on code availability)

Reviewer #2

(Remarks to the Author)

In this manuscript, the authors have conducted integrated metabolite profiling, transcriptomics, and photophysiological analyses on the early land plant *Physcomitrium patens*, along with the streptophyte algae *Zygnema circumcarinatum* and *Mesotaenium endlicherianum*. This study is anchored on two pivotal terrestrial environmental factors: temperature and light. Employing co-expression analysis, time-series transcriptome analysis, and gene regulatory network inference, the authors have delineated significant regulatory networks responsive to environmental and apocarotenoid signals. The findings offer valuable insights, suggesting that an ancient web of kinase hubs, where environmental and apocarotenoid signals converged in the last common ancestor of land plant and streptophyte algae. Although the study provides lots of information and has the potential to make contributions to plant terrestrialization, the logic of the whole study is vague, and the most important finding of this work is less prominent. The authors need to re-shape the logic and extensively expand the discussion.

Major points:

1. It seems that the authors try to emphasize on "convergence" during the whole manuscript, however, it is still difficult to easily understand the importance of "convergence" after reading through the manuscript.
2. In line 115: The average RNA-seq mapping rate for Zc appears to be relatively low. Are there any outliers in the mapping rates for some samples, and does this affect the reliability of the results and conclusions? I suggest to provide a supplementary table, which lists the conditions and the corresponding RNA-seq mapping rate for each sample.
3. In Fig. 2a, the separation between treatment conditions for Zc is not as distinct when compared to the other two species, especially since the cold and control conditions overlap for the most part. Could this ambiguity stems from the data quality issues, or does it reflect actual biological phenomenon?
4. In the section "Dynamic xanthophyll cycle and apocarotenogenesis in Mesotaenium,": why focus on Mesotaenium? It seems to deviate from the main focus of the study, which examines the concerted action of land plants with streptophyte algae? Could the authors provide explanations about how this focus aligns with or contributes to the broader narrative of the study?
5. Although I appreciate the wonderful work on retracing regulatory networks responsive to environmental and apocarotenoid signals, I think the full discussion of the implications of this work is obviously missing. For instance, what novel insights into plant terrestrialization are provided by this time-series transcriptome-based analyses compared to the previous work by the authors. The current "DISCUSSION" is more likely as the "CONCLUSION".
5. The figure 2-5 should be simplified and consolidated by including only key information and results. To be specific: (1) The line graph of principal component analysis (Fig. 2a) might be placed in the supplemental figure. (2) Figure 3b, c, and e exhibit the same issues; one of them should be retained while the others can be placed in the supplemental figure or all of them should be re-integrated. (3) The information provided by Figure 4c and d is limited, and the multiple network in Figure 4e-j are homogenized in terms of their forms; therefore, it is recommended to include them in the supplemental figure or re-integrated. (4) The information of Figure 5c is insufficient, making it difficult for the readers to find useful information from it.

Other points:

1. In lines 71-72: "A physicochemical consequence of the elevation of atmospheric oxygen levels due to plant terrestrialization and radiation³⁸ might have been higher rates in apocarotenogenesis - independent of the evolution of carotenoid-cleaving enzymes." I am not sure if the second half of the above sentence is entirely correct, but it might be helpful to include a citation here.
2. In line 88: The article on *Zygnema circumcarinatum* has been published and is now accessible online, please cite the published version.
3. In Fig. 1c, the presentation of the mathematical model equations is difficult to discern. Kindly revise the formatting for improved clarity and readability.
4. In Fig. 2, figure notes do not correspond to statements made in the manuscript; for instance, Fig. 2c is not labeled. Please make the necessary corrections.
5. In lines 142-143: The statement mentions that "Changes in the relative xanthophyll pool size were within $\pm 20\%$ upon 2 hours of treatment by any stressor relative to t_0 ". However, the distinctions in pigment pool alterations in Fig. 3c are not readily apparent. To enhance the clarity of the illustration, it is recommended to improve the presentation. For instance, incorporating numerical labels to highlight the changes would improve the readability of the figure.
6. In lines 146-147: "Since enzymatic reactions of the carotenoid pathway are favored by heat and other biosynthetic steps are favored by light in both conditions". Please include a citation here.
7. Please add a summary diagram to illustrate ancient kinase hubs in the last common ancestor of embryophytes and algae in an intuitive manner, while also providing a detailed description and discussion.

(Remarks on code availability)

Reviewer #3

(Remarks to the Author)

The manuscript by Rieseberg et al. presents a well-written and comprehensive study on the adaptation mechanisms of the earliest land plants to environmental stressors. The authors effectively combine a variety of methodologies, including photophysiology, transcriptomics, and metabolite profiling, to explore stress response kinetics across three divergent streptophytes. This manuscript offers a wealth of new data and insights into the ancient gene regulatory networks that enabled these plants to adapt to the dynamic stress conditions encountered on land.

While the study is thorough and the findings are interesting, I have several comments and suggestions that I will outline

below to further strengthen the manuscript:

Major comments:

1. While the images presented are undoubtedly complex and visually striking, I question whether all of them are necessary. It seems that many of the main figures could be significantly simplified to better support and emphasize the key conclusions outlined in the discussion section (which are largely missing, see major comment 2). Aside from Figure 6, none of the network visualizations are legible in print, nor are they essential for understanding the data. Other issues with data visualization include microscopic and uninformative row and column labels, overlapping clusters marked in similar colors like yellow, and some less effective choices in data representation (specific cases are detailed in my minor comments).
2. Considering the extensive data presented, the discussion section feels quite limited, focusing primarily on the conclusion that the connectivity of the molecular network of interest is largely mediated by kinases. This conclusion, though important, seems somewhat underwhelming given the complexity and scale of the analysis performed. It raises the question of whether this finding was not expected, and whether it could have been reached through much less complex methods, such as differential gene expression analysis. Expanding the discussion to include a more comprehensive summary of all conclusions related to "oxidative signal convergence across the algae-embryophyte divide" based on the data and performed analyses would significantly enhance the reader's understanding and appreciation of the study's broader implications.

Minor and Specific Comments:

3. Lines 50-53 - The sentence "Yet, we are only beginning to understand how these genes might have been used in a functional advantage at the time of the conquest of land" is unclear. Which genes are referred to? No specific genes were mentioned earlier, only traits.
4. In the section "Global transcriptomics bear out divergent time-course dynamics," the authors refer to Fig. 2a, b, and c, but the figure only contains panels a and b. I assume that panel c in the text refers to panel b in the figure?
5. **Figure 2b (referred to as c in the text)** - This figure is very difficult to understand, likely for a wider audience unfamiliar with such data representation. What does the y-axis of the plots represent? How were the listed GO terms selected, and how is their order along the y-axis determined? What does it mean when they change order or width? There is no explanation provided in the legend or the main text. Please improve clarity.
6. **Figure 3c** - The pie charts here do not seem like the best choice. It is unclear how the pigments are comparable in terms of their pool sizes. What is the biological or biochemical meaning of the pie size? Does the total pie size represent some absolute unit? I strongly suggest using bar plots with clearly defined units.
7. Lines 145-146 - The statement "This is also reflected by transcript level changes in homologs of enzymes likely responsible for gross flux regulation (Fig. 3d)" requires a citation as no flux measurements were performed.
8. Lines 175-176 - The wording is unclear in "We then asked the questions of how these clusters (i) correlate with the environmental cues and apocarotenoid levels (Fig. 4b) and (ii) are similar across species (Fig. 4c,d)." Clusters cannot correlate. Do you mean the average expression profile of the clusters?
9. Lines 188-191 - It seems none of the referred figures show the correlation with temperature. What exactly do the reported r values refer to? Are the average profiles of clusters correlated with the vector of temperature for each sample? Please clarify.
10. Line 211 - The term "focal occurrence" is unclear.
11. **Figure 5b** - Why are so many clusters colored in yellow? Very confusing.
12. **Figure 5c** - This plot is very complex and not intuitive; it takes time to understand. The legend is not helpful, and the axis labels are microscopic even when zoomed in on a screen. Axes are not labeled. It is hard to imagine this will be visible in print. Is this figure necessary?
13. Line 221 - The text mentions that 24 metabolites were used in the network generation, but their predicted connectivity with genes and the conservation/specificity of this connectivity is not sufficiently detailed in the manuscript. It would be highly valuable to evaluate whether the integrated network makes biological sense.
14. **Paragraph: "Temporal stress co-expression and Granger causal inference of gene regulatory networks"** - It is unclear to what degree the functional annotation of genes as transcription factors or DNA-binding proteins is taken into account in the GRN inference. How was the performance of the GRN reconstruction benchmarked? Is there evidence for true regulator-target interactions from orthologs, or is it based solely on GO annotation of network modules? Clear evidence that the presented GRN contains more true interactions than a shuffled control is needed.
15. Line 225 - Is the number of 923 conserved gene pairs large or small relative to the number of genes and possible gene pairs used in the inference?
16. Lines 230-233 - This part is very interesting and important for the manuscript and requires more statistical analysis: For instance, are hubs more conserved than non-hubs? Is the directionality significantly conserved or random? What indicates that the network contains regulatory interactions rather than just reflecting another form of data clustering?
17. Lines 341-342 - The phrase "Our findings pinpoint the hubs in which this information is bundled" is unclear. What does "information bundling" mean?

(Remarks on code availability)

Reviewer #4

(Remarks to the Author)

This manuscript shows extensive time-resolved profiling of three related photosynthetic species to explore the dynamics of stress response. The main observations of this work based on integration of photophysiology datasets, transcriptomics, and pigment profiling reveal the network of oxidative stress response across the three streptophyte species, which are connected

through kinase signaling hubs. The interspecies patterns are the most interesting, having identified patterns in the oxidative stress response and ethylene signaling across divergent species. Although I find the analysis to be intriguing, I think the readability of this manuscript for this journal could be improved given the complexity of the dataset and methodology. In particular, the figures are incredibly dense and a bit overwhelming. Most of my suggestions are related to better communication of the findings through the figures.

Here are several comments/suggestions for the authors to address:

- In Figure 2a, the equations for the fits of the principal components are not very legible and may not be necessary. And what is the y-axis in the alluvial plots in Figure 2b? I also think that 2b and 2c are labeled differently from what is referenced in the text.
- Figure 3b caption states, "Heatmap of metabolite flux and its correlation". Flux implies a kinetic rate. Can the authors clarify what measurement is being shown?
- In Figure 4f-j, the cluster networks don't reveal much structure. Did they look at different cutoffs for visualization? Because of the interconnectedness of the networks, it's challenging to read the gene names. Are there overlaps between the hub genes identified in the clusters of the different species - for example, the protein homeostasis clusters? Is Figure 4k referred to in the text? The caption could also be clearer.
- Figure 5b is also confusing with regards to which clusters were shown. The clusters Pp C16 and Me C1 don't appear to be correlated in expression at all, and to a lesser extent, Pp C16 and Me C2 also look distinct. Why do these clusters have a high similarity coefficient?
- It's hard to follow what is being shown and the significance of the DPGP clusters in Figure 5c based on the main text description. It's a very detailed figure, but barely discussed.
- The conserved patterns inferred from the gene regulatory network are interesting, especially through the kinase hubs. I wonder if the authors can comment on the species-specific differences in signaling and regulation, especially since they observe divergent responses to some of the stresses in their profiling experiments?

(Remarks on code availability)

The code is well organized and readable. The repository also includes the data files and plots, which is very useful for the community. The authors did a great job putting this resource together along with the interactive Shiny app.

Version 2:

Reviewer comments:

Reviewer #1

(Remarks to the Author)

I thank the authors for their efforts to answer all of my concerns and make the analysis more accessible. I have no further questions.

(Remarks on code availability)

Reviewer #2

(Remarks to the Author)

I appreciate the feedback and revisions provided by the authors. The revised manuscript has been significantly improved in various aspects. I have only a few of points for authors' consideration.

1) The emphasis in Fig. 3b is not as pronounced as desired. With the aim of enhancing the article's readability, it could be more fitting to consider it as a supplementary illustration. Nevertheless, this is not a fundamental issue, and I defer to the author's ultimate choice. For Fig. 3c, if a pie chart is indeed necessary, please add a legend to indicate the values corresponding to the different sizes of the entire pie charts, as it has been noted that the authors have intentionally used pie charts of different sizes to convey information.

2) I recommend that the Fig. 7a and b could be rotated 90 degrees clockwise for easier viewing. Alternatively, also considering to place these figures in the appendix.

3) Fig. 8 serves as a summary chart and is very concise and clear. However, please check if the image is complete as it appears to be truncated at the bottom.

(Remarks on code availability)

Reviewer #3

(Remarks to the Author)

The authors have successfully addressed most of my concerns, and the manuscript is much improved. I have just a few

minor points:

1. Lack of time units: In Figure 2 b and c, and Figure 3 d, please include time units on the x-axes and in the color keys.
2. The use of phrase “expression behavior”: While the context makes sense, the phrase “expression behavior” might confuse some readers (cognitive science?). “Expression pattern” might be more precise.
3. Comment 12: The revisions for Comment 12 look good, and I agree it works best as a supplemental figure. On the side note, concerning the “While it is all nicely visible in print (we tested this for every figure)”, no it is not! And I will die on this hill together with our printer.

(Remarks on code availability)

The code looks ok and annotated, but im missing a README with some overview.

Reviewer #4

(Remarks to the Author)

The authors have addressed my concerns in their revised version.

(Remarks on code availability)

RESPONSE TO REVIEWER COMMENTS

We would like to thank all reviewers for the very constructive feedback. While the additional requests prompted the generation of 8 all-new supplemental figures and several new sub-panels, we made sure to strike a balance with streamlining some of the figures and making all the material more accessible. Thanks to the comments of the reviewers and the changes they prompted, we believe that our manuscript has further improved.

Reviewer #1 (Remarks to the Author):

In this study, Rieseberg et al. generated extensive transcriptomic and metabolomic datasets across three streptophytes that diverged 600 million years ago to study stress response kinetics. Through network analysis, the authors discovered that major kinases and ethylene signaling components predate ancient convergent signaling pathways. I would like to express my appreciation for their efforts in providing many valuable and high-quality resources for the community. Additionally, both the analytical pipeline and the bioinformatic analysis were well-designed. Overall, this study is novel and has significant merits for various research areas. However, the manuscript could benefit from some reorganization of the text and data analysis, as well as data visualization. >>>>AU: We thank the reviewer for appreciating the scope and analyses that went into our work. Further, we would like to thank the reviewer for the input that stimulated some additional analyses and made the manuscript overall more accessible.

Major points:

The descriptions in the results section are overly descriptive. The authors' claims would be stronger if they presented their data in a more quantitative manner.

>>>>AU: We have worked on adding more statements that rest on quantifications of the patterns that we have observed in the data, including for remarks by other reviewers. This includes (1) clearly stating the rank of the hubs, (2) quantification of the degree distribution of the network that characterise it as scale-free, (3) a quantification of the authority of the hubs versus non-hub genes as well as their degrees, (4) add more quantitative information on gene relationships to orthogroups, and (5) we now show that most of the top100 hubs are recurrently recovered among the highest rank in the predicted GRNs, no matter whether the full gene or gene plus metabolite dataset is used for computing the SWING-RF network (see the all-new quantification in Figure 7c).

1. Line 76-77: I didn't find the data correlating time-resolved transcriptome and metabolomic measurements. The stated correlation between traits (metabolite contents) and gene expression (modules) does not clearly reflect the kinetics of time-resolved data.

>>>>AU: Maybe this was a bit unclear. Indeed, we do several approaches that in the end are based on calculating correlations. Foremost, we now specify that we calculated correlations. The result of such calculations can of course be that things are not correlated (some are, some are not). The predicted GRN presented in Figure 6 is based on a correlative method (Granger causality), which is only possible because of time-resolved data—the SWING methods builds on these. If the reviewer is asking for a correlation between individual genes and the pigment levels (maybe we are reading too much between the lines), we have provided this as an overview figure (all-new supplemental Figure 17) — but it is extremely complex and there is (to us) no straightforward pattern.

2. Fig 1c: The kinetics in Pp appear to differ significantly from the other two algae and are particularly sensitive to temperature shifts, especially during heat stress. Can the authors provide an explanation for this? Also, what are the cell numbers after 24 hours? Is the recovery of Fq'/Fm' in Pp at 24 hours related to this?

>>>>AU: Well-spotted. We have explored this aspect a bit further based on finding groups of temporally clustered genes (Jaccard similar) that show a distinct temporal behaviour in *Physcomitrium*. We find that *Physcomitrium* apparently responds with an early upregulation of genes enriched for functions in multicellular developmental processes (links of morphogenesis, cell-cell interaction, and stress response). Especially given that *Physcomitrium* has the most complex body of the organisms investigated here, this distinct response could govern the differences in heat response. This stands in contrast to the morphological differences (of which we observed none) and thus there is no clear connection between growth phenotypes and Fq'/Fm'. We have added this information to the paragraphs where we describe the DPGP results and created an all-new supplemental integrated into the old Figure S6 (see Suppl. Figure 6c, 6d, 6e). All of these additional analyses (including all comparisons made, not only the ones highlighted in the all-new supplement) are now also provided online on GitHub.

3. Line 115: Given the recent publication of a high-quality whole genome in the three species, can the authors explain the lower mapping rate in Zc?

>>>>AU: There are several aspects to consider. The assuring aspect is that we see that the mapping rate is consistent with the mapping rate that was also obtained for the data in the *Zygnema* genome paper — and this held true across strains and conditions for *Zygnema*. In our opinion, this consistently (!) lower mapping rate it points to some RNA processing / RNA types we cannot currently account for by directly mapping onto the genome. In Dadrás et al. 2023 we already observed the major impact that RNAseq data garnered from diverse conditions can have on the predictive power for finding genes in Zygnematophyceae. As it currently stands, we have the least diverse (conditions) RNAseq data on *Zygnema*, whereas both *Physcomitrium* and *Mesotaenium* are much more researched organisms. Thus, these

numbers simply highlight that once additional data on *Zygnema* were garnered (this work is a good first step in that direction), it is a worthwhile effort to re-run ab initio gene model predictions—but this goes far beyond the scope of this study.

For the purpose of this work, we have now provided an additional plot in the supplemental Methods (previously called Supplemental Material), showing that 85 out of 90 samples are within the $\pm 5\%$ range of the average mapping rate, hence excluding the possibility that there is a bias due to the treatments.

4. Line 118: I do not see the support for the claim that “HL showed swift changes in all three species”. The separation of PC1 for HL in Pp is not evident.

>>>>AU: Good point. We have specified this that only heat has this swift and pronounced effect and HL only for the two algae.

5. Extended Data Fig 2: Pp seems to recover faster. Can the authors comment on this observation? It also appears that the kinetics differ significantly among the three species according to the PCA analysis. Additionally, the GO analysis suggests that different terms are enriched in the three species. Please clarify in the text.

>>>>AU: Thanks for this good observation. Indeed, it is fully consistent with the Fv/Fm assessment. We have added this information to the text. Regarding the GO analysis, see our response to your remark number 7.

6. Line 130, Extended Data Fig 3: I can't locate the terms mentioned in the figure. It would be helpful if the authors could highlight these terms directly in the figure.

>>>>AU: We have now highlighted the cohorts of terms mentioned in the text with a yellow background color.

7. Fig. 2b: Does the GO enrichment pattern remain consistent when only orthologous genes are used for analysis? Are the observed differences primarily due to orthologous or non-orthologous genes? Furthermore, the authors combine three different types of GO terms (BP, MF, and CC) but only discuss BP terms in the main text. The clarity of the figure could be improved by focusing solely on BP functions.

>>>>AU: Thank you for the stimulating suggestion. We have replotted the alluvial plot solely based on BP terms. Overall, the take home message remains very similar and the plot is equally informative. We thus created an all-new supplemental Figure 2c that contains the BP version; we prefer to keep the plot that includes all three GO term categories because in our view it provides the most holistic bird's eye view of the data. The first part of your query was very revealing also for us. Indeed, we find that irrespective of the treatment, more than 50% of differentially expressed genes fall into the same HOG (see the all-new supplemental Figure 2a).

8. Line 130: Could the authors provide additional details about the enzymes in each species? Specifically, information on how many orthologs exist in each species, along with their locus ID numbers and gene names, would be useful.

>>>>AU: Sure—good point. To make it more accessible, we have provided all the enriched GO terms, including the number of genes associated to it, as an all-new supplemental table 1. Please note that the information is also available via the interactive web interface provided by our Shiny app, which we created so that any GO term of interest (along with all genes predicted to be assigned to that ontology) can be extracted from the dataset.

9. Fig 3b: The authors use the term "flux" in the legend; however, they only measure the abundance changes at different time points of the treatments. I'm unsure if "flux" is the correct term to use in this context.

>>>>AU: Very good point. We remove the flux statement and wording.

10. Line 140, Fig 3d: The term "relevant gene" needs a clearer definition in the main text. Additionally, the legend states "one respective homolog shown"; how do the authors select which homolog to display in the figure? What about the other homologous genes in each species? Also, how are the genes in Fig 3d related to those in Fig 3a? Are they all key enzymes shown in Fig 3a?

>>>>AU: Point well-taken. For an overview, we have generated the same plot as a supplementary item (the all-new Suppl. Figures 14 to 16) that shows the expression dynamics of all homologs detected.

11. Lines 146-147: It is unclear which enzymes or pathways are favored by which stress conditions based on these figures. Additionally, the gene expression trends in Fig 3d for high light (HL) and heat stress (HS) appear quite similar. Could the authors clarify this point?

>>>>AU: Good point. We have modified and simplified the statement.

12. Line 160: The results here indicate no correlation between metabolites and gene expression, whereas Line 77 in the introduction mentions a correlation. The authors should provide clearer evidence to support the correlation between these data.

>>>>AU: See our response above to number one. Just to clarify this misunderstanding. In line 77 we did not mean that there was correlation but that we calculated (i.e. performed correlation analyses) between all the different lines of data.

13. Line 179: It's unclear from Fig. 4d; perhaps the authors could highlight the relevant parts of the figure. Additionally, marking the genes within the network could enhance clarity.

>>>>AU: We have added labels to the two highlighted examples and marked the genes in the network.

14. Fig. 4: How many genes are in each module? Do modules with similar functions from different species contain comparable numbers of genes or homologous gene group (HOG) numbers? Are the similar GO terms derived from modules based on the analysis of orthologs or all genes?

>>>>AU: We now show all gene numbers next to any WGCNA-derived module (Figure 4, Suppl. Fig. 5) and created an all-new supplemental table 2 with the gene numbers.

We thank the reviewer from prompting this meta-analysis: The analysis of GO terms enriched among similar clusters are shown in an all-new Suppl. Fig. 18. Indeed, recurrent patterns in response regulation were recovered.

15. How do the authors select hub genes for display in the networks?

>>>>AU: These are the 20 most connected genes. We have modified the figure legend to specify that these are the hubs: "...the top 20 most connected genes (the hubs) annotated based on homology..."

16. Line 187: Zc appears to be markedly different. Can the authors provide an explanation for this?

>>>>AU: Absolutely. We have added this now already in the section that also features the explanation of Figure 2 (because here it is immediately apparent for the first — but interestingly not last [pointing to the very exiting biology of Zygnema] — time); we now explain: "This subdued response of Zc (here and in the following) is likely due to its natural ecophysiology and growth, allowing the formation of highly resilient algal mats that even withstand the harsh environments of the High Arctic." and refer to Rippin et al. 2019.

17. Line 206: Why are there different numbers of clusters in each species? Which number of clusters did the authors use in the formal analysis?

>>>>AU: There are different numbers of clusters because clustering was done by condition, i.e. for each species there are four categories of clusters. We now specify this in the text and show the number of clusters before and after the Gaussian probability cutoff of 0.7 in the new supplemental table 2. Furthermore, since this was apparently leading to some confusion, we have generated a workflow (Suppl. Method Fig. 14) and an extensive accompanying text.

18. Fig. 5 & Fig. 6: The authors predict several novel genes involved in the convergent signaling network and infer numerous potential regulations among these interactions. However, the entire section (Lines 202-324) reads more like a discussion rather than a presentation of results.

>>>>AU: Thanks a lot for appreciating the relevance of our analysis. We have restructured the entire text to have a "Results and Discussion" section, ending with a whole new general discussion followed by a formal "Conclusion" section.

19. Discussion Section: In the time-course Gene Regulatory Network (GRN) analysis, the authors argue that kinases form a network, acting as convergence points between inputs and outputs. However, the kinetics of these enzymes appear to vary significantly among the three species, and the dynamics of metabolites also differ. It would enhance clarity if the authors could link Fig. 5 to Fig. 3 and highlight the critical enzyme versus metabolite changes in Fig. 6.

>>>>AU: Thank you for this suggestion. The temporal dynamics are indeed quite complex (e.g. the new Suppl. Fig. 17), while showing often clear correlation. This is one of the reasons why we have moved beyond simply working with the DPGP-derived temporal clusters but have used advanced methodology with SWING-RF. We have now however used several measures to draw better connections between the different analyses and highlight the conserved set that can be discerned from the data (new Figures 4l, 7, Suppl. Fig. 6c-e). To make it better accessible we have added more discussion points and a summary Figure (all-new Figure 8).

Minor points:

1. There is no Fig. 2c mentioned in the manuscript.

>>>>AU: Well-spotted. 2b were supposed to be the line plots and 2c the alluvial plots. It is corrected now.

2. Line 152: Fig. 3e does not depict gene expression.

>>>>AU: Corrected to 3d.

3. What does the grey line (or grey area) represent in Fig. 5c?

>>>>AU: Upon request of another reviewer, we now moved this to Extended Data Figure 6 and now explain in the caption that "Grey lines mark those DPGP clusters whose expression profiles are shown on the right."

4. There is no Fig. 5d in the manuscript.

>>>>AU: This must have been a misunderstanding. We looked up the mentioning of a 5d and we do not refer to (a main) Fig. 5d but to Extended Data Figure 5d.

5. Fig. 6: What is the rationale behind selecting these particular hubs for display?

>>>>AU: For the nodes in the network: this is the entire network after all the filtering steps (we outline them more clearly now and better contextualized in an expanded explanation in the methods); there was no additional selection.

When it comes to the labelling, we added the explanation that “Hierarchical orthogroups (HOGs) are annotated if they engage in the top 100 most predictive relationships”.

6. The figure legends could be more precise, particularly the descriptions in the Gene Regulatory Network (GRN) sections, such as those in Lines 734 and 828.

>>>>AU: We have re-read all the figure legends and modified them for clarity.

7. Lines 831 & 833: It appears that some words may be missing.

>>>>AU: We have fixed this.

Reviewer #2 (Remarks to the Author):

In this manuscript, the authors have conducted integrated metabolite profiling, transcriptomics, and photophysiological analyses on the early land plant *Physcomitrium patens*, along with the streptophyte algae *Zygnema circumcarinatum* and *Mesotaenium endlicherianum*. This study is anchored on two pivotal terrestrial environmental factors: temperature and light. Employing co-expression analysis, time-series transcriptome analysis, and gene regulatory network inference, the authors have delineated significant regulatory networks responsive to environmental and apocarotenoid signals. The findings offer valuable insights, suggesting that an ancient web of kinase hubs, where environmental and apocarotenoid signals converged in the last common ancestor of land plant and streptophyte algae. Although the study provides lots of information and has the potential to make contributions to plant terrestrialization, the logic of the whole study is vague, and the most important finding of this work is less prominent. The authors need to re-shape the logic and extensively expand the discussion.

>>>>AU: Thanks a lot for appreciating the scope of our work. We have restructured the entire text to have a “Results and Discussion” section, ending with a whole new general discussion followed by a formal “Conclusion” section. For the details see our response to your major point 5.

Major points:

1. It seems that the authors try to emphasize on “convergence” during the whole manuscript, however, it is still difficult to easily understand the importance of “convergence” after reading through the manuscript.

>>>>AU: We have now added a quantification of the authority of the hubs versus non-hub genes as well as their degrees.

2. In line 115: The average RNA-seq mapping rate for *Zc* appears to be relatively low. Are there any outliers in the mapping rates for some samples, and does this affect the reliability of the results and conclusions? I suggest to provide a supplementary table, which lists the conditions and the corresponding RNA-seq mapping rate for each sample.

>>>>AU: There are several aspects to consider. The assuring aspect is that we see that the mapping rate is consistent with the mapping rate that was also obtained for the data in the *Zygnema* genome paper — and this held true across strains and conditions for *Zygnema*. In our opinion, this consistently (!) lower mapping rate it points to some RNA processing / RNA types we cannot currently account for by directly mapping onto the genome. In Dadras et al. 2023 we already observed the major impact that RNAseq data garnered from diverse conditions can have on the predictive power for finding genes in *Zygnematophyceae*. As it currently stands, we have the least diverse (conditions) RNAseq data on *Zygnema*, whereas both *Physcomitrium* and *Mesotaenium* are much more researched organisms. Thus, these numbers simply highlight that once additional data on *Zygnema* were garnered (this work is a good first step in that direction), it is a worthwhile effort to re-run *ab initio* gene model predictions—but this goes far beyond the scope of this study.

For the purpose of this work, we have now provided an additional plot in the supplemental Methods (previously called Supplemental Material), showing that 85 out of 90 samples are within the $\pm 5\%$ range of the average mapping rate, hence excluding the possibility that there is a bias due to the treatments.

3. In Fig. 2a, the separation between treatment conditions for *Zc* is not as distinct when compared to the other two species, especially since the cold and control conditions overlap for the most part. Could this ambiguity stem from the data quality issues, or does it reflect actual biological phenomenon?

>>>>AU: We are convinced that this is a biological phenomenon because it is consistent with the changes in metabolites (where *Zygnema* shows the smallest), published data that shows *Zygnema*'s resilience (e.g., doi: 10.1111/pp1.14056), and what we know about *Zygnema*'s ecophysiology. To explain this, we added the statement “This subdued response of *Zc* (here and in the following) is likely due to its natural ecophysiology and growth, allowing the formation of highly resilient algal mats that even withstand the harsh environments of the High Arctic.” and refer to Rippin et al. 2019. Further, after long treatment treatment, we do see gene expression changes popping up. If it were a data problem, we would not see these. *Zygnema* simply has a slower temporal progression.

4. In the section “Dynamic xanthophyll cycle and apocarotenogenesis in *Mesotaenium*,”: why focus on *Mesotaenium*? It seems to deviate from the main focus of the study, which examines the concerted action of land

plants with streptophyte algae? Could the authors provide explanations about how this focus aligns with or contributes to the broader narrative of the study?

>>>>AU: The narrative in the text is about all species. This section title was simply to highlight that *Mesotaenium* showed the most dynamics. To better emphasize our intention of highlighting the unique response patterns of *Mesotaenium*, we changed the sub-heading to "Xanthophyll cycle and apocarotenogenesis were the most dynamic in *Mesotaenium*".

5. Although I appreciate the wonderful work on retracing regulatory networks responsive to environmental and apocarotenoid signals, I think the full discussion of the implications of this work is obviously missing. For instance, what novel insights into plant terrestrialization are provided by this time-series transcriptome-based analyses compared to the previous work by the authors. The current "DISCUSSION" is more likely as the "CONCLUSION".

>>>>AU: We have taken several measures to address this point. (1) We have now highlighted that our results are a "Results and Discussion" section (because a lot of the discussion was already in it), (2) Turned the previous "Discussion" into a proper formal "Conclusion", (3) added a completely new two-paragraph discussion section to the end of Results and Discussion, and (4) a new conceptual figure (the all-new Figure 8). One however needs to keep in mind that we were already above 4000 words in the main text so we have to make sure to stay concise.

5. The figure 2-5 should be simplified and consolidated by including only key information and results. To be specific: (1) The line graph of principal component analysis (Fig. 2a) might be placed in the supplemental figure. (2) Figure 3b, c, and e exhibit the same issues; one of them should be retained while the others can be placed in the supplemental figure or all of them should be re-integrated. (3) The information provided by Figure 4c and d is limited, and the multiple network in Figure 4e-j are homogenized in terms of their forms; therefore, it is recommended to include them in the supplemental figure or re-integrated. (4) The information of Figure 5c is insufficient, making it difficult for the readers to find useful information from it.

>>>>AU: Thank you for the helpful suggestions, which we considered carefully. We have modified all figures mentioned to make them more accessible and integrated them better into the flow of the text. For figure 2a we have now relabelled the panel as 2b and kept it in the main figure because we now state more clearly what the purpose of this was (we actually think that the line graph is the most accessible and straightforward way to see the main dynamics). Since the different panels in Figure 3 show very different things, we did not really see a good way of removing one without creating a gap of information in the flow of the text. That being said, we have done two things: (i) we have better integrated them into the flow of the main text and (ii) provided all-new supplemental figures 13 to 16 to have a different view at the data. Figure 4c is important in our opinion to understand how one goes to 4d. In 4d we now added highlights and also in figures 4f to 4j we now have highlighted more genes of interest. We further highlight in the figure legend now that "All networks can be explored interactively at <https://rshiny.gwdg.de/apps/streptotime>." Figure 5c was now moved to the supplement and we now present a much slimmer version of Figure 5.

Other points:

1. In lines 71-72: "A physicochemical consequence of the elevation of atmospheric oxygen levels due to plant terrestrialization and radiation³⁸ might have been higher rates in apocarotenogenesis - independent of the evolution of carotenoid-cleaving enzymes." I am not sure if the second half of the above sentence is entirely correct, but it might be helpful to include a citation here.

>>>>AU: This is our thought / speculation. There is no citation.

2. In line 88: The article on *Zygnema circumcarinatum* has been published and is now accessible online, please cite the published version.

>>>>AU: Corrected.

3. In Fig. 1c, the presentation of the mathematical model equations is difficult to discern. Kindly revise the formatting for improved clarity and readability.

>>>>AU: We have removed these in order to streamline the figures.

4. In Fig. 2, figure notes do not correspond to statements made in the manuscript; for instance, Fig. 2c is not labeled. Please make the necessary corrections.

>>>>AU: Corrected.

5. In lines 142-143: The statement mentions that "Changes in the relative xanthophyll pool size were within $\pm 20\%$ upon 2 hours of treatment by any stressor relative to t_0 ". However, the distinctions in pigment pool alterations in Fig. 3c are not readily apparent. To enhance the clarity of the illustration, it is recommended to improve the presentation. For instance, incorporating numerical labels to highlight the changes would improve the readability of the figure.

>>>>AU: We have plotted the data in a more accessible format and provide this in the supplement (the all-new Suppl. Figure 13). For the main figure, we would like to keep the pie charts because they are in our view much more intuitive than the bar charts.

6. In lines 146-147: "Since enzymatic reactions of the carotenoid pathway are favored by heat and other biosynthetic steps are favored by light in both conditions". Please include a citation here.
>>>>AU: **In the process of making the text more accessible this part of the sentence is gone and the sentence was wholly restructured.**

7. Please add a summary diagram to illustrate ancient kinase hubs in the last common ancestor of embryophytes and algae in an intuitive manner, while also providing a detailed description and discussion.
>>>>AU: **A very nice suggestion that we gladly followed. We have added a summary diagram as Figure 8. See also our response to your major point 5.**

Reviewer #3 (Remarks to the Author):

The manuscript by Rieseberg et al. presents a well-written and comprehensive study on the adaptation mechanisms of the earliest land plants to environmental stressors. The authors effectively combine a variety of methodologies, including photophysiology, transcriptomics, and metabolite profiling, to explore stress response kinetics across three divergent streptophytes. This manuscript offers a wealth of new data and insights into the ancient gene regulatory networks that enabled these plants to adapt to the dynamic stress conditions encountered on land.

While the study is thorough and the findings are interesting, I have several comments and suggestions that I will outline below to further strengthen the manuscript:

>>>>AU: **Thank you appreciating the value of our work and the constructive feedback. We have taken all into account and responded to your remarks point-by-point below.**

Major comments:

1. While the images presented are undoubtedly complex and visually striking, I question whether all of them are necessary. It seems that many of the main figures could be significantly simplified to better support and emphasize the key conclusions outlined in the discussion section (which are largely missing, see major comment 2). Aside from Figure 6, none of the network visualizations are legible in print, nor are they essential for understanding the data. Other issues with data visualization include microscopic and uninformative row and column labels, overlapping clusters marked in similar colors like yellow, and some less effective choices in data representation (specific cases are detailed in my minor comments).

>>>>AU: **We have made sure to increase the accessibility of the figures through several measures. The details are outlined in the responses below.**

2. Considering the extensive data presented, the discussion section feels quite limited, focusing primarily on the conclusion that the connectivity of the molecular network of interest is largely mediated by kinases. This conclusion, though important, seems somewhat underwhelming given the complexity and scale of the analysis performed. It raises the question of whether this finding was not expected, and whether it could have been reached through much less complex methods, such as differential gene expression analysis. Expanding the discussion to include a more comprehensive summary of all conclusions related to "oxidative signal convergence across the algae-embryophyte divide" based on the data and performed analyses would significantly enhance the reader's understanding and appreciation of the study's broader implications.

>>>>AU: **We have, also in response to the other reviewer, expanded the discussion. The reason was that we have initially tried to keep the discussion very concise and focused on what is at hand. We have now added a completely new discussion section and have drawn a conclusion figure.**

The conclusion that we draw was—at least to us—not expected at all. We were in fact very surprised by it. We would not have thought that kinase-coding genes are among those that predicted to be transcriptionally (!) highly influenced by various genes. In fact, we would have thought that anything around kinases mainly happens on a posttranslational level. Furthermore, several other surprises were uncovered, including the role of subtilase/peptide signaling (which we thus highlight a bit better in the text), strong bolstering of the ethylene/light quality track and more. Overall this pushes several exiting aspects of shared streptophyte biology into the limelight—and we hope it will provide an impetus for the field that goes beyond the current state-of-the-art.

Minor and Specific Comments:

3. Lines 50-53 - The sentence "Yet, we are only beginning to understand how these genes might have been used in a functional advantage at the time of the conquest of land" is unclear. Which genes are referred to? No specific genes were mentioned earlier, only traits.

>>>>AU: **Good point. We have driven the point home that these traits are underpinned by complex genes.**

4. In the section "Global transcriptomics bear out divergent time-course dynamics," the authors refer to Fig. 2a, b, and c, but the figure only contains panels a and b. I assume that panel c in the text refers to panel b in the figure?

>>>>AU: **Indeed. We have added the missing panel label.**

5. **Figure 2b (referred to as c in the text)** - This figure is very difficult to understand, likely for a wider audience unfamiliar with such data representation. What does the y-axis of the plots represent? How were the listed GO terms selected, and how is their order along the y-axis determined? What does it mean when they change order or width? There is no explanation provided in the legend or the main text. Please improve clarity.

>>>>AU: Indeed, this is an unusual representation but one that we find very intuitive for time-course data. But we do fully agree that our explanation was quite meagre. We have modified the figure legend, it now specifies: "Time-resolved alluvial diagrams of the most enriched GO terms found across at least three time points based on significantly differentially regulated genes. Different terms are separated by color. The enrichment of a term is indicated by the width of the line. Enrichment analyses were performed by comparing the treatments cold, heat and high light stress exposure versus control. The x-axis represents the time scale. Arrangement of terms along the y-axis shows which is, at a given time point, at which rank among the most enriched."

6. **Figure 3c** - The pie charts here do not seem like the best choice. It is unclear how the pigments are comparable in terms of their pool sizes. What is the biological or biochemical meaning of the pie size? Does the total pie size represent some absolute unit? I strongly suggest using bar plots with clearly defined units.

>>>>AU: We can understand how the pie charts seem an unusual choice but this is actually not uncommon in the field presenting the pool of the xanthophyll pigments. The pie charts are scaled according to the pool size. We considered switching these to bar plots as suggested but found that they are far less intuitive than the pie charts. However, to provide some more such standard plots, we have also added the bar plots to the supplement.

7. Lines 145-146 - The statement "This is also reflected by transcript level changes in homologs of enzymes likely responsible for gross flux regulation (Fig. 3d)" requires a citation as no flux measurements were performed.

>>>>AU: Good point. We have completely modified this sentence and also now fully avoid the phrase "flux".

8. Lines 175-176 - The wording is unclear in "We then asked the questions of how these clusters (i) correlate with the environmental cues and apocarotenoid levels (Fig. 4b) and (ii) are similar across species (Fig. 4c,d)." Clusters cannot correlate. Do you mean the average expression profile of the clusters?

>>>>AU: Absolutely correct. Thanks. We have modified the phrasing.

9. Lines 188-191 - It seems none of the referred figures show the correlation with temperature. What exactly do the reported r values refer to? Are the average profiles of clusters correlated with the vector of temperature for each sample? Please clarify.

>>>>AU: The correlation is what is shown in Figure 4b. To make this more accessible, we have now uploaded all detailed plots on the correlation on Zenodo under: [10.5281/zenodo.14234484](https://zenodo.org/record/10.5281/zenodo.14234484).

10. Line 211 - The term "focal occurrence" is unclear.

>>>>AU: Thank you for bringing this to our attention. We have reworded this sentence completely.

11. **Figure 5b** - Why are so many clusters colored in yellow? Very confusing.

>>>>AU: Because we keep a consistent coloring code. This yellow is throughout the manuscript for HL, as in Figures 1, 2, 3, and 5.

12. **Figure 5c** - This plot is very complex and not intuitive; it takes time to understand. The legend is not helpful, and the axis labels are microscopic even when zoomed in on a screen. Axes are not labeled. It is hard to imagine this will be visible in print. Is this figure necessary?

>>>>AU: Good point. While it is all nicely visible in print (we tested this for every figure) we agree that it is a very complicated figure for a relatively simple message. We have thus moved it to the supplement for exploring it on a screen. We find it an interesting idea to do this comparison (and find it quite useful for exploring the data) but we fully agree that it was a bit much for a simple message.

13. Line 221 - The text mentions that 24 metabolites were used in the network generation, but their predicted connectivity with genes and the conservation/specificity of this connectivity is not sufficiently detailed in the manuscript. It would be highly valuable to evaluate whether the integrated network makes biological sense.

>>>>AU: See our response to 14.

14. **Paragraph: "Temporal stress co-expression and Granger causal inference of gene regulatory networks"** - It is unclear to what degree the functional annotation of genes as transcription factors or DNA-binding proteins is taken into account in the GRN inference. How was the performance of the GRN reconstruction benchmarked? Is there evidence for true regulator-target interactions from orthologs, or is it based solely on GO annotation of network modules? Clear evidence that the presented GRN contains more true interactions than a shuffled control is needed.

>>>>AU: Thank you for inquiring more about this because it is the foundation of our analysis. Since there have been a couple of questions regarding the approach, we have added a Supplementary Method Figure 14 that shows the workflows for these analyses. Further, we have added a detailed rationale of the network-based approaches.

The idea behind SWING is to compare time-series prediction models to infer Granger causality. It uses only and only time-series gene expression profile, and receives the size of the window (of time points) to calculate Granger causality via a windowed strategy over time. During the modeling, each gene will be treated as a response variable, while other genes (at different time delays) serve as explanatory variables. Here, user should define the minimum and maximum time delays that are meaningful based on the biological question at hand and experiment design. Then, using a Random Forest algorithm determines feature importance scores for each window of time using a multivariate Granger causality (via calculating the mean squared error). An adjacency matrix will be created for each window where a_{ij} is the inferred score for gene i regulating gene j . The rank of an edge in each windowed model can be used as the confidence metric to compare across methods. SWING does two aggregations. Confidence values from windowed subsets are aggregated into a single network by taking the mean rank of the edge at each delay k , and then taking the mean rank of the edge across all delays. The window specific models (aka adjacency matrices) are aggregated to create a consensus network by aggregating edge ranks across windows and delays and computing the mean and median for each edge. The output will be a directed edge list (aka graph) that shows regulatory relationship predicted via this method and their confidence scores. Multivariate Granger causality evaluates whether past values of a gene improve the prediction of another gene. It also incorporates multiple explanatory genes and time delays. We used the Random forest option with 500 ensemble of decision trees to assess Granger causality in time-series gene expression data. Permutation testing is also used to establish null distribution for feature importance scores. By randomly shuffling explanatory variables, SWING can evaluate whether observed scores significantly exceed what is expected by chance or not. The edges that are not consistently ranked highly across multiple windows are removed during the aggregation step. Furthermore, we would like to stress that the hubs we identify in the SWING-based prediction of the GRN also show up as hubs in the WGCNA-based unsupervised clustering; on the robustness of the hubs, see our response to your query 16.

15. Line 225 - Is the number of 923 conserved gene pairs large or small relative to the number of genes and possible gene pairs used in the inference?

>>>>AU: Good point—this was not contextualized so far. We now refer to the methods, where we added the explanation: “Based on the input, theoretically possible network could consist of 3,596,712, 13,641,942, and 2,652,012 edges for the networks of *Me*, *Pp*, and *Zc*, respectively. However, we recovered networks with 3,557,677 (98.9147%), 3,910,293 (28.66376%), and 1,415,350 (53.36891%) non-zero predictive pairs of genes (*Me*, *Pp*, and *Zc*, respectively). To calculate the conserved network, we mapped gene IDs to HOGs to find a common concept across species. The number of edges (the predictive pairs) are 1,465,716, 1,246,870, 856,719 for *Me*, *Pp*, and *Zc*, respectively. We had 5552 HOGs that are shared between *Zc*, *Me*, and *Pp*. Therefore, in theory the maximum number of predictions (pairs of genes) is 30,819,152. The network that was conserved among *Zc*, *Pp*, and *Me* had 383,984 (1.245927%) unique pairs. Finally, we added an extra layer of filtering by just looking at the genes that were among conserved DPGP clusters across species with probability bigger than 0.7. Here, we are looking at genes that are both tightly clustered across species (DPGP) and have conservation of clustering across species (Orthofinder and Jaccard distance). Using these as filters on top of the SWING method, significantly reduce the number of genes, but it is the set of genes that are show tight regulation at expression level as well as homology besides the Granger causality. Here, we had 923 HOGs and metabolites that were present in conserved DPGP clusters. Therefore, in theory, we could have maximum 851,006 edges in the conserved network. We had 923 edges (0.1084599%). In summary, the conserved gene pairs (923) filtered based on similar clusters of co-expression networks (DPGP) and homology (Orthofinder) are a very small portion of the potential/theoretical network—obtained after applying several filters for conservation and robustness.” As outlined in response to your query 13/14, several measures that assure the robustness of the predictions via SWING-RF were applied.

16. Lines 230-233 - This part is very interesting and important for the manuscript and requires more statistical analysis: For instance, are hubs more conserved than non-hubs? Is the directionality significantly conserved or random? What indicates that the network contains regulatory interactions rather than just reflecting another form of data clustering?

>>>>AU: Thank you for this important point. We have now added a quantification of the authority of the hubs versus non-hub genes as well as their degrees, showing that the hubs stand out in an all-new inset in Figure 6. As outlined in the response to your query 13/14, the predictions of Granger causality between genes/metabolites is filtered based on multiple statistical cutoffs. Thus, by definition, it is not random. Thanks for raising the important point that some readers might equate Granger causality between genes/metabolites with direct biological interaction. In order to avoid this confusion, we have added the sentence: “Note that the predicted GRN has been reconstructed based on Granger causality which is not the same as biological regulatory interactions; it rather reflects whether gene expression changes in one gene are a significant explanatory factor in the expression change of another gene.” Finally, we now show that most of the top100 hubs are recurrently recovered among the highest rank in the predicted GRNs, no matter whether the full gene or gene plus metabolite dataset is used for computing the SWING-RF network (see the all-new quantification in Figure 6g).

17. Lines 341-342 - The phrase "Our findings pinpoint the hubs in which this information is bundled" is unclear. What does "information bundling" mean?

>>>>AU: It is rephrasing the idea that hubs are a point of convergence. With the inset in Figure 6a we have now provided

quantitative data on this aspect. Also see the new quantification on the hub ranks in the all-new Figure 7c.

Reviewer #4 (Remarks to the Author):

This manuscript shows extensive time-resolved profiling of three related photosynthetic species to explore the dynamics of stress response. The main observations of this work based on integration of photophysiology datasets, transcriptomics, and pigment profiling reveal the network of oxidative stress response across the three streptophyte species, which are connected through kinase signaling hubs. The interspecies patterns are the most interesting, having identified patterns in the oxidative stress response and ethylene signaling across divergent species. Although I find the analysis to be intriguing, I think the readability of this manuscript for this journal could be improved given the complexity of the dataset and methodology. In particular, the figures are incredibly dense and a bit overwhelming. Most of my suggestions are related to better communication of the findings through the figures.

>>>>AU: Thank you for appreciating the value of our work and for the constructive suggestions. We have worked with all of the remarks and think that this way the accessibility of our study improved.

Here are several comments/suggestions for the authors to address:

- In Figure 2a, the equations for the fits of the principal components are not very legible and may not be necessary. And what is the y-axis in the alluvial plots in Figure 2b? I also think that 2b and 2c are labeled differently from what is referenced in the text.

>>>>AU: Indeed. We have fixed the labeling and (to make the figures less busy) we have removed the equations.

- Figure 3b caption states, "Heatmap of metabolite flux and its correlation". Flux implies a kinetic rate. Can the authors clarify what measurement is being shown?

>>>>AU: Very good point. We remove the flux statement (also in response to another reviewer).

- In Figure 4f-j, the cluster networks don't reveal much structure. Did they look at different cutoffs for visualization? Because of the interconnectedness of the networks, it's challenging to read the gene names. Are there overlaps between the hub genes identified in the clusters of the different species - for example, the protein homeostasis clusters?

>>>>AU: The clusters are quite large, the focus here is to give examples on the hub genes. We have now made sure that they are more readable in print. Exploring the overlaps between hub genes revealed a very interesting pattern indeed! We found and write that "Overall, the 1860 top20 hub genes found among the total of 93 clusters computed by WGCNA for all three species belonged to 1473 orthogroups. Of these, 1047 (71.1%) were orthogroups containing hubs in all three species (Fig. 4l), indicating their shared high connectivity across 600 million years of divergent evolution". This is shown in the all-new Fig 4l.

Is Figure 4k referred to in the text? The caption could also be clearer.

>>>>AU: This is mentioned more clearly now.

- Figure 5b is also confusing with regards to which clusters were shown. The clusters Pp C16 and Me C1 don't appear to be correlated in expression at all, and to a lesser extent, Pp C16 and Me C2 also look distinct. Why do these clusters have a high similarity coefficient?

>>>>AU: This is actually the beauty of the analysis. The high similarity is based on similarity in gene occurrence, not simply based on a similar topology of the expression dynamics. This way, related genes that behave similar and different can be found.

- It's hard to follow what is being shown and the significance of the DPGP clusters in Figure 5c based on the main text description. It's a very detailed figure, but barely discussed.

>>>>AU: Good point. Also in response to another reviewer, we agree that it is a very complicated figure for a relatively simple message. We have thus moved it to the supplement.

- The conserved patterns inferred from the gene regulatory network are interesting, especially through the kinase hubs. I wonder if the authors can comment on the species-specific differences in signaling and regulation, especially since they observe divergent responses to some of the stresses in their profiling experiments?

>>>>AU: Good point. We have added an analysis that the genes in the major kinase hub show extremely dynamic gene expression patterns, adding a whole-new suppl. Figure 11. In line with the comments above about the particular responsiveness of the organisms (particularly of algae to HL and the moss to temperature), we now added "The expression of these Ser/Thr kinase-coding genes were highly dynamic across species, in particular fast spikes early upon HL in the algae and temperature in Pp (Suppl. Fig. 11) reflecting the physiological responsiveness."

Reviewer #4 (Remarks on code availability):

The code is well organized and readable. The repository also includes the data files and plots, which is very useful for the community. The authors did a great job putting this resource together along with the interactive Shiny app.

>>>>AU: Thank you.

RESPONSE TO REVIEWER COMMENTS

We would like to thank again all reviewers for the very constructive feedback.

REVIEWERS' COMMENTS

Reviewer #1 (Remarks to the Author):

I thank the authors for their efforts to answer all of my concerns and make the analysis more accessible. I have no further questions.

>>>>AU: Thank you.

Reviewer #2 (Remarks to the Author):

I appreciate the feedback and revisions provided by the authors. The revised manuscript has been significantly improved in various aspects. I have only a few of points for authors' consideration.

1) The emphasis in Fig. 3b is not as pronounced as desired. With the aim of enhancing the article's readability, it could be more fitting to consider it as a supplementary illustration. Nevertheless, this is not a fundamental issue, and I defer to the author's ultimate choice. For Fig. 3c, if a pie chart is indeed necessary, please add a legend to indicate the values corresponding to the different sizes of the entire pie charts, as it has been noted that the authors have intentionally used pie charts of different sizes to convey information.

>>>>AU: We think it is important to provide this figure to make sure that a bird's eye overview of the pigment data are provided. We have added a key for the pie chart figure and the pool size values.

2) I recommend that the Fig. 7a and b could be rotated 90 degrees clockwise for easier viewing. Alternatively, also considering to place these figures in the appendix.

>>>>AU: Good point, we rotated panels a and b.

3) Fig. 8 serves as a summary chart and is very concise and clear. However, please check if the image is complete as it appears to be truncated at the bottom.

>>>>AU: We checked again and all appears correct. The nucleus and the DNA therein are supposed to emerge like this.

Reviewer #3 (Remarks to the Author):

The authors have successfully addressed most of my concerns, and the manuscript is much improved. I have just a few minor points:

1. Lack of time units: In Figure 2 b and c, and Figure 3 d, please include time units on the x-axes and in the color keys.

>>>>AU: Added.

2. The use of phrase "expression behavior": While the context makes sense, the phrase "expression behavior" might confuse some readers (cognitive science?). "Expression pattern" might be more precise.

>>>>AU: Good point. We changed this.

3. Comment 12: The revisions for Comment 12 look good, and I agree it works best as a supplemental figure. On the side note, concerning the "While it is all nicely visible in print (we tested this for every figure)", no it is not! And I will die on this hill together with our printer.

>>>>AU: Thank you. Point well taken.

Reviewer #3 (Remarks on code availability):

The code looks ok and annotated, but im missing a README with some overview.

>>>>AU: We have added a README file.

Reviewer #4 (Remarks to the Author):

The authors have addressed my concerns in their revised version.

>>>>AU: Thank you.